# Aromatic side-chain conformational switch on the surface of the RNA Recognition Motif enables RNA discrimination

Nana Diarra dit Konté [1], Miroslav Krepl[2,3], Fred F. Damberger[1], Nina Ripin[1], Olivier Duss[1,4,5], Jiří Šponer[2,3] & Frédéric H.-T. Allain[1]

The cyclooxygenase-2 is a pro-inflammatory and cancer marker, whose mRNA stability and translation is regulated by the CUG-binding protein 2 interacting with AU-rich sequences in the 3′ untranslated region. Here, we present the solution NMR structure of CUG-binding protein 2 RRM3 in complex with 5′-UUUAA-3′ originating from the COX-2 3′-UTR. We show that RRM3 uses the same binding surface and protein moieties to interact with AU- and UG-rich RNA motifs, binding with low and high affinity, respectively. Using NMR spectroscopy, isothermal titration calorimetry and molecular dynamics simulations, we demonstrate that distinct sub-states characterized by different aromatic side-chain conformations at the RNA-binding surface allow for high- or low-affinity binding with functional implications. This study highlights a mechanism for RNA discrimination possibly common to multiple RRMs as several prominent members display a similar rearrangement of aromatic residues upon binding their targets.

[1] Department of Biology, ETH Zürich, Institute of Molecular Biology and Biophysics, HPP L 14.1 Hönggerbergring 64, 8093 Zürich, Switzerland. [2] Institute of Biophysics, Academy of Sciences of the Czech Republic, Kralovopolska 135, 612 65 Brno, Czech Republic. [3] Department of Physical Chemistry, Faculty of Science, Regional Centre of Advanced Technologies and Materials, Palacky University Olomouc, 17.listopadu 12, 771 46 Olomouc, Czech Republic. [4] Department of Integrative Structural and Computational Biology, The Scripps Research Institute, MB-33 10550 North Torrey Pines Road, La Jolla, California 92037, USA. [5] Department of Structural Biology, Stanford University, Stanford, CA 94305, USA. Correspondence and requests for materials should be addressed to F.H.-T.A. (email: frederic.allain@mol.biol.ethz.ch)

Messenger RNA processing is an important landmark in gene regulation and a plethora of RNA-binding proteins controls and modulates its different steps acting as essential components of ribonucleoparticles. Understanding protein-RNA recognition mechanisms in general and the interaction of the ubiquitous RNA recognition motif (RRM) protein domains with their cognate targets in particular is essential to gain insight in posttranscriptional gene regulation. RRMs are the most abundant RNA-binding domain[1]. They are characterized by two conserved ribonucleoprotein (RNP) consensus sequences; RNP1 and RNP2 composed of aromatic and charged residues interspersed by hydrophobic ones. RRMs are about 90 amino acids long and adopt a βαββαβ secondary structure forming an antiparallel β-sheet stacked onto two α-helices[1]. Canonically, they bind two to three nucleotides on the β-sheet through the RNPs. RNA-binding proteins such as CUG-BP2 contain multiple RRM copies and are involved in several regulatory processes. To perform successfully its various functions, CUG-BP2 recognizes numerous RNA targets. The protein binds to CUG triplet repeats[2], but SELEX experiments identified the UG dinucleotide and the 5′-UGUU-3′ sequence to be bound preferentially[3]. These motifs are found in several mRNA targets of CUG-BP2, notably in the cystic fibrosis transmembrane conductance regulator mRNA[3, 4]. UV-cross linking and gel shift assays established that AU-rich sequences are also targets of CUG-BP2[5, 6]; these include the AU-rich sequence found upstream of the edited cytosine of apolipoprotein B mRNA and the adenine- and uridine-rich elements (AREs) mostly localized in the 3′-untranslated region (UTR) of unstable mRNAs[7]. The latter are 50–150 nucleotide long sequences composed of $AU_nA$ repeats. They associate with ARE-binding proteins such as CUG-BP2 that can positively or negatively modulate mRNA stability and translation levels. CUG-BP2 stabilizes the mRNA of the cyclooxygenase 2 (COX-2), which is a regulatory enzyme of prostaglandins metabolism but inhibits its translation. COX-2 is overexpressed in epithelial malignancies, where it plays a role in tumorigenesis and protection against damage due to γ-radiation[5]. In response to apoptotic stimuli[8, 9], the cytoplasmic concentration of CUG-BP2 increases thus down-regulating COX-2 and thereby favoring apoptosis. Owing to the variety of its functions and binding partners, studying the binding mode of CUG-BP2 to its multiple RNA sequence targets can help elucidate not only its role in ARE-mediated mRNA decay but also how proteins, and particularly the ubiquitous RRM-containing proteins discriminate various RNA targets.

CUG-BP2 belongs to the "CUG-BP1 and ELAV type RNA-binding protein 3-like factor" (CELF) family. All six members share structural and sequence similarities but CUG-BP1 and CUG-BP2 are the most closely related with over 90% identity in conserved regions. CELF proteins contain three RRMs (Fig. 1a); the two RRMs located in the N-terminal region are separated from the third RRM by a low complexity region called the divergent domain, which is thought to be involved in protein-protein interactions.

Structures of the two N-terminal RRMs of CUG-BP1 bound to UG-rich RNAs containing the 5′-UGUU-3′ motif and of the third RRM bound to 5′-UGUGUG-3′ have been determined[10, 11]. The available crystal structures reveal that RRM1 and RRM2 bind the 5′-UGUU-3′ motif similarly. At the β-sheet RNA-binding interface of both domains, the UG dinucleotide shares structural features with the left-handed Z-RNA helix and only the last two uracils stack with the conserved phenylalanines of the RNPs. All four nucleotides are specifically recognized by hydrogen-bonding to the protein backbone and side-chains. In contrast, the solution structure of CUG-BP1 RRM3 in complex with $(UG)_3$ shows that four out of six nucleotides stack on aromatic residues of the β-sheet surface and specific recognition is ensured by a dense network of hydrogen-bonds between RNA moieties and the protein backbone and polar side-chains. Owing to the high degree of identity between CUG-BP1 and CUG-BP2 RRMs, these structures provide very detailed insight into the binding mode to UG-rich RNAs for both proteins.

To understand how CUG-BP2 recognizes AU-rich RNAs and how the same RRM can recognize diverging targets (UG-rich and AU-rich RNAs), we focused our investigations on the C-terminal RRM, which has a particularly high density of aromatic residues at the binding interface. In addition to the three canonical aromatic residues of the RNP1 and RNP2, two phenylalanines and one histidine are exposed on the β-sheet surface. We here determined the solution structure of RRM3 bound to 5′-UUUAA-3′ RNA revealing that the RRM3 uses the same protein moieties to recognize the AU-rich and $(UG)_3$[11] RNAs. However, the aromatic side-chains undergo conformational rearrangement when bound to 5′-UGUGUG-3′, whereas they remain in a conformation similar to the free protein in presence of 5′-UUUAA-3′. The two complexes differ in affinity but mutational studies revealed an unexpected affinity increase for both types of RNA sequences upon replacement of different aromatic residues by alanine. Scalar coupling measurements and molecular dynamics (MD) simulations demonstrated that the binding surface of CUG-BP2 RRM3 exists in multiple states and that a conformational switch of aromatic side-chains is at the origin of the fine tuning of the affinity for different targets.

## Results

### RRM3 binds to AU-rich motifs found in the COX-2 mRNA 3′-UTR.
To elucidate the recognition of AREs by CUG-BP2, we investigated the interaction between AU-rich RNAs and the C-terminal RRM3. We titrated RRM3 with 5′-AUUUAAUU-3′, a sequence from the COX-2 mRNA 3′-UTR. Upon addition of the oligonucleotide to RRM3 at 40 °C, we observed protein chemical shift changes in the 2D $^1$H–$^{15}$N HSQC indicating RNA binding (Fig. 1b), albeit in the fast exchange regime relative to the NMR time scale (Supplementary Fig. 1). Analysis of the 2D $^{13}$C-half-filtered NOESY of $^{13}$C, $^{15}$N-labeled RRM3 in complex with unlabeled 5′-AUUUAAUU-3′ in $D_2O$ revealed severe overlap and similar intermolecular NOE patterns for equivalent resonances from consecutive RNA nucleotides, which indicated the presence of several binding registers on the protein and prevented a structure determination of this complex (see $A_5$ and $A_6$ resonances in Supplementary Fig. 2a). To reduce the exchange between multiple registers, we titrated a shorter oligonucleotide 5′-UUUAA-3′ to RRM3. We also lowered the temperature to 25 °C to reduce the rates of any remaining exchange phenomena. RRM3 binding to 5′-UUUAA-3′ causes chemical shift changes of smaller amplitude than with 5′-AUUUAAUU-3′ (Fig. 1b). Nonetheless, the same residues are affected and the amide resonances move in similar direction, suggesting a similar binding mode for the two RNAs (Fig. 1c). $K_d$ values of RRM3 for 5′-AUUUAAUU-3′ and 5′-UUUAA-3′ at 25 °C determined by NMR titrations, showed a two-fold decrease in affinity upon shortening the RNA (Fig. 1d, Table 1). However, the improved spectral quality with the pentamer, where no register exchange was observed (Supplementary Fig. 2b), allowed the structure determination of CUG-BP2 RRM3 bound to 5′-UUUAA-3′.

### Solution NMR structure of CUG-BP2 RRM3/UUUAA.
We adopted an approach previously described for weak affinity complexes[12] to solve the structure of CUG-BP2 RRM3 bound to 5′-UUUAA-3′ (Table 2). We derived 2325 intra-protein distance restraints from NOESY experiments carried out on samples with

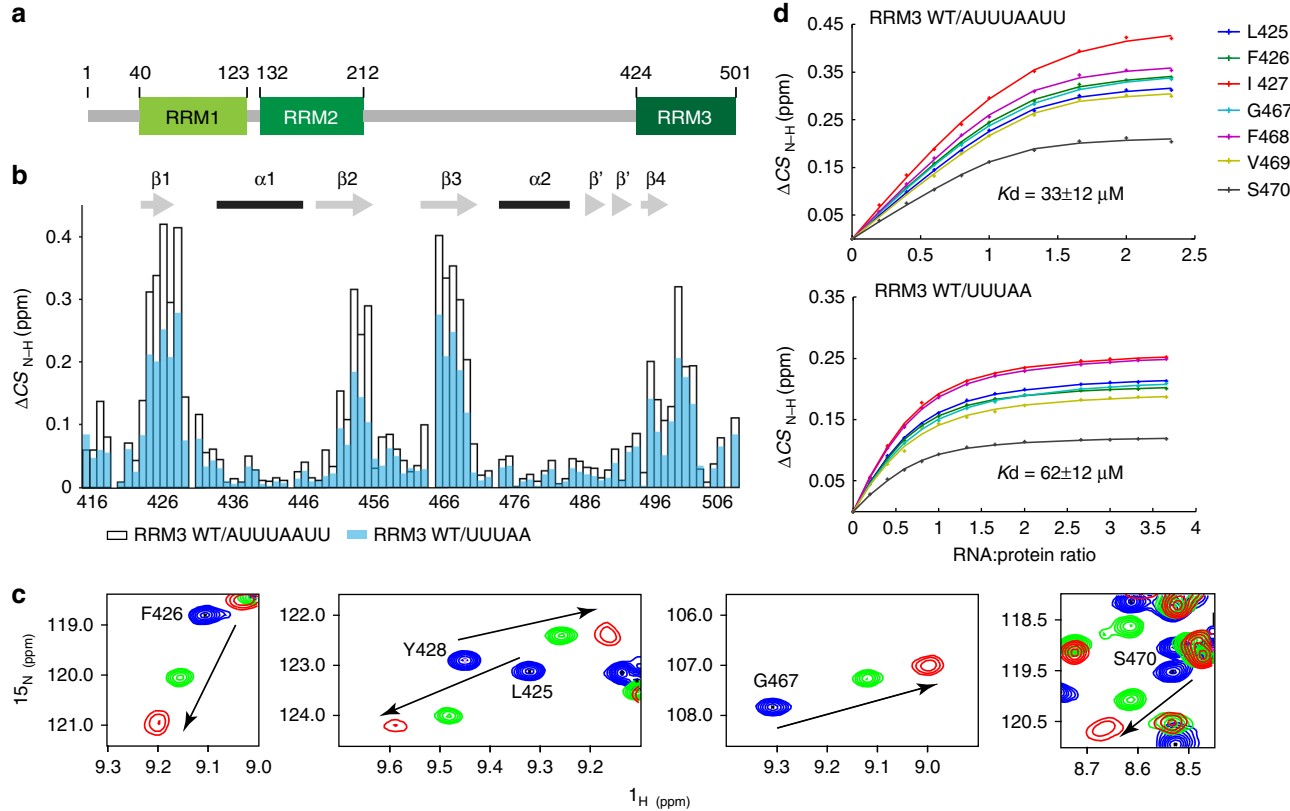

**Fig. 1** Interaction of CUG-BP2 RRM3 with AU-rich RNAs. **a** Schematic representation of full-length CUG-BP2. **b** Mapping of the combined chemical shift perturbations (ΔCS) observed on the backbone NH signals of RRM3 upon addition of saturating amounts of 5′-AUUUAAUU-3′ (1:1.33, *open bars*) and 5′-UUUAA-3′ (1:3, *blue bars*). **c** Overlaid expansions of the $^1$H-$^{15}$N 2D-HSQCs of RRM3 alone (*blue*) and in the presence of saturating amounts of 5′-AUUUAAUU-3′ (*red*) or 5′-UUUAA-3′ (*green*). **d** Titration curves of CUG-BP2 RRM3 NH signals with 5′-AUUUAAUU-3′ (*top*) and 5′-UUUAA-3′ (*bottom*), respectively at 25 °C

a 1- to 3- protein/RNA ratio, where 96% of RRM3 is present in the bound form. Although this ensures that an insignificant amount of free protein is detected by NMR, most of the RNA signals arise from the free form. Therefore, the 49 intermolecular and the 56 RNA intramolecular distance restraints were derived from NOESY spectra measured on samples with a 1-to-1 protein to RNA ratio. Under these conditions, the free unstructured RNA yields only weak NOEs and the intermolecular NOEs arise from bound RNA. Using this strategy, we could calculate 50 conformers satisfying nearly all the input constraints. The 20 conformers with the lowest energy after refinement were selected to constitute the final ensemble, which displayed an RMSD of 0.67 Å for the heavy atoms in the structured regions (Fig. 2a, Table 2). CUG-BP2 RRM3 adopts the typical RRM fold (βαββαβ)[1] and similar to CUG-BP1 RRM3, the N-terminus is ordered and lies across the β-sheet surface[11].

**Recognition mode of UUUAA RNA.** The 5′-UUUAA-3′ RNA occupies a positively charged cleft of RRM3 that is delimited by the β-sheet, the N- and C-termini and the β2–β3 loop (Fig. 2b). The aromatic residues exposed at the interface provide the main binding platform (Fig. 2c) and together with the peripheral polar side-chains, they delimit 5 binding pockets N-1, N0, N1, N2 and N3 (using the nomenclature defined by Auweter et al[13], Fig. 3a, b). In the N-1 pocket, the $N_\varepsilon$–$H_\varepsilon$ moiety of the protonated His429 is hydrogen-bonded to either the base O2 or the ribose O2′ of $U_1$ (Fig. 3b and Supplementary Table 1). The latter hydrogen-bond is observed in many of the conformers with the ribose of $U_1$ in a C3′ endo conformation whereas the remaining sugars always have a

C2′ endo pucker. Tyr428 together with the canonical aromatic residues Phe426 and Phe468 are involved in π–π stacking with the base moieties of $U_2$, $U_3$ and $A_4$, respectively. Phe455 makes hydrophobic contacts with the sugar moiety of $A_5$. Additional non-specific interactions are provided by the main-chain amide of His429, which forms a hydrogen-bond with a phosphate oxygen of $U_2$, the side-chain of Arg500 with $U_3$ O2′, the amide of Ile 456 with $A_5$ O2′ and the side-chain amino group of Lys464 which is hydrogen-bonded with $A_5$ O3′.

In addition to these nonspecific interactions, each base of the pentanucleotide is recognized by at least one hydrogen-bond to the base. The side-chain of Lys495 is positioned between the bases of the first two nucleotides and its amino group is hydrogen-bonded either to $U_1$ O2 or $U_2$ O4. A similar configuration is observed on the other side of the binding interface where the Lys453 amino group is hydrogen-bonded either to the N3 of $A_4$ or $A_5$. In the N1 pocket, the recognition of $U_3$ is typical of a uracil binding at this position in RRMs. The presence of a hydrogen-bond between U3 O2 and the Arg500 main-chain amide is supported by its distinctive down-field chemical shift (Supplementary Fig. 1). An interesting feature of this complex is the triad formed by Glu418, Lys499 and $A_4$ in the pocket N2. The side-chain carboxyl of the Glu418 forms a salt bridge with the Lys499 side-chain amine and is hydrogen-bonded to $A_4$ amino. Finally, in the N3 pocket, the $A_5$ N1 atom is hydrogen-bonded to the Gln416 side-chain amide.

**RRM3 plasticity in RNA recognition.** The present structure of CUG-BP2 RRM3 in complex with 5′-UUUAA-3′ along with the

**Table 1 List of $K_d$ values determined by NMR $^{15}$N-HSQC[a] or ITC[b] titrations**

| Protein | RNA (5′–3′) | $K_d$ (μM) | $K_d$ ratio | Resonances[c] |
|---|---|---|---|---|
| RRM3 wild-type[a] | AUUUAAUU | 31 ± 9 | 2.2 | 10 |
| | UUUAA | 66 ± 14 | - | 6 |
| | **C**UUAA | 54 ± 9 | 1.2 | 7 |
| | U**C**UAA | 76 ± 27 | 0.9 | 8 |
| | UU**C**AA | 56 ± 12 | 1.2 | 8 |
| | UUU**C**A | 73 ± 13 | 0.9 | 12 |
| | UUUA**C** | 56 ± 8 | 1.2 | 13 |
| RRM3 wild-type | UUUAA[a] | 62 ± 12 | - | 6 |
| F426A | | 139 ± 10 | 0.3 | 6 |
| Y428A | | 27 ± 3 | 2.5 | 8 |
| H429A | | 30 ± 4 | 2.2 | 12 |
| K453A | | 78 ± 13 | 0.9 | 8 |
| F455A | | 41 ± 5 | 1.6 | 15 |
| K495A | | 63 ± 6 | 1.1 | 12 |
| Q497A | | 60 ± 10 | 1.1 | 5 |
| K499A | | 91 ± 18 | 0.7 | 7 |
| RRM3 wild-type | UGUGU[b] | 1.04 ± 0.04 | - | |
| Y428A | | 0.31 ± 0.01 | 3.3 | |
| H429A | | 0.27 ± 0.05 | 3.3 | |
| F455A | | 10.39 ± 0.87 | 0.1 | |

All $K_d$ values were determined by NMR titration except for the complexes with 5′-UGUGU-3′ that were determined by ITC. All $K_d$ ratio are relative to the $K_d$ of RRM3 wild-type in complex with 5′-UUUAA-3′ except for the complexes with 5′-UGUGU-3′ that are relative to the $K_d$ of RRM3 wild-type in complex with 5′-UGUGU-3′
[c]Number of resonances in 2D $^{15}$N HSQC used to calculate the average $K_d$

**Table 2 Structural statistics for CUG-BP2 RRM3/5′-UUUAA-3′ complex**

| | |
|---|---|
| *NMR distance and dihedral constraints* | |
| Distance constraints | 2452 |
| RRM3 total NOE | 2325 |
| Intra-residue | 473 |
| Inter-residue | 1852 |
| Sequential (li-jl = 1) | 654 |
| Medium range (1 < li-jl < 5) | 462 |
| Long range (li-jl > = 5) | 736 |
| Hydrogen bonds[a] | 22 |
| UUUAA total NOE | 56 |
| Intra-residue | 34 |
| Inter-residue | 22 |
| Sequential (li-jl = 1) | 22 |
| Medium range (1 < li-jl < 5) | 0 |
| Long range (li-jl > = 5) | 0 |
| Hydrogen bonds[a] | 0 |
| Complex intermolecular | 49 |
| Hydrogen bonds[a] | 0 |
| Total dihedral angle restraints | 135 |
| Phi | 67 |
| Psi | 63 |
| Sugar pucker (DELTA) | 5 |
| *Structure statistics*[b] | |
| Average number of distance constraint violations | |
| 0.1–0.2 Å | 45.0 ± 5.1 |
| 0.2–0.3 Å | 10.1 ± 3.1 |
| 0.3–0.4 Å | 1.9 ± 1.2 |
| > 0.4 Å | 0.6 ± 0.6 |
| Maximal distance violation (Å) | 0.41 ± 0.06 |
| Average angle constraint violations | |
| < 5 ° | 6.2 ± 1.0 |
| > 5 ° | 0.6 ± 0.5 |
| Maximal angle violation (°) | 7.02 ± 4.74 |
| Mean Deviation from ideal covalent geometry | |
| Bond Length (Å) | 0.0037 ± 0.0001 |
| Bond angle (°) | 1.976 ± 0.014 |
| *Ramachandran plot statistics*[b,c,d] | |
| Residues in most favored regions (%) | 91.8 ± 1.5 |
| Residues in additionally allowed regions (%) | 8.2 ± 1.5 |
| Residues in generously allowed regions (%) | 0.0 ++ 0.0 |
| Residues in disallowed regions (%) | 0.0 ± 0.0 |
| *RMSD to mean structure statistics*[b,c] | |
| RRM3 | |
| Backbone atoms | 0.28 ± 0.04 |
| Heavy atoms | 0.58 ± 0.05 |
| UUUAA | |
| Backbone atoms | 0.77 ± 0.19 |
| Heavy atoms | 0.90 ± 0.17 |
| Complex | |
| Backbone atoms | 0.43 ± 0.06 |
| Heavy atoms | 0.67 ± 0.06 |

[a]Each hydrogen bond is defined by two restraints (H - Acceptor and Donor - Acceptor)
[b]Statistics calculated for the deposited bundle of 20 structures. Values are reported as mean ± SD
[c]Residue range: 24–105 for RRM3 and 1–5 for RNA
[d]Ramachandran statistics evaluated by PROCHECK-NMR.32

CUG-BP1 RRM3/5′-UGUGUG-3′ complex solved by Tsuda et al.[11] provide detailed structural information on how RRM3 adapts its binding mode to the two sequences. The strict sequence conservation at the binding interface in the third RRM of the two proteins allows for a direct comparison of their interactions with 5′-UGUGUG-3′ and 5′-UUUAA-3′. The two RNAs occupy the same binding surface with the pockets N0, N1, N2 and N3 being common to both complexes (Fig. 3b, c). The same protein moieties are involved in making intermolecular contacts, however, the aromatic side-chain conformations at the binding interface differ strikingly between the two complexes (Fig. 3d). The structure presented here reveals that the aromatic residues contacting 5′-UUUAA-3′ have conformations that resemble those observed in the solution structure of the free CUG-BP1 RRM3[11] (Fig. 4b). In what follows, we define this state as the UP conformation since the aromatic residues are pointing "up" relative to the β-sheet when displayed in the standard RRM orientation. The higher affinity target 5′-UGUGUG-3′ is bound on the RRM with many of the key aromatic side-chains pointing "down"; we therefore refer to this state as the DOWN conformation.

We probed the sequence-specific nature of the complex by scanning mutagenesis of 5′-UUUAA-3′ RNA. Replacing individually each nucleotide by a cytosine does not change the apparent affinity despite the loss of a few sequence-specific contacts (Table 1, Supplementary Fig. 3). The $K_d$ values extracted from titrations of RRM3 with the RNA mutants monitored by $^{15}$N-HSQCs do not differ significantly from the value of 62 ± 12 μM obtained for the wild-type RNA, suggesting that the UP-binding mode represents at best a complex with low sequence-specificity. The chemical-shift changes of the protein when comparing complexes of the wild-type and mutant RNAs do not cluster around the mutated base, indicating that there is an adaptation of the entire binding interface to the change in sequence (Supplementary Fig. 4). The specific contacts lost due to the base mutation are either compensated by new interactions with the functional moieties of the cytosine or by a shift of binding

register of the RNA. RRM3 displays clear adaptability when interacting with different uridine-rich RNAs. However, it has a strong binding preference for 5′-UGUGUG-3′. The $K_{ds}$ determined by NMR for CUG-BP2 RRM3 in complex with 5′-UUUAA-3′ or the cytosine-containing RNA mutants are all about 60 μM, which is 30-fold higher than the $K_d$ obtained by Tsuda et al. for the CUG-BP1 RRM3/5′-UGUGUG-3′ complex (1.9 μM)[11].

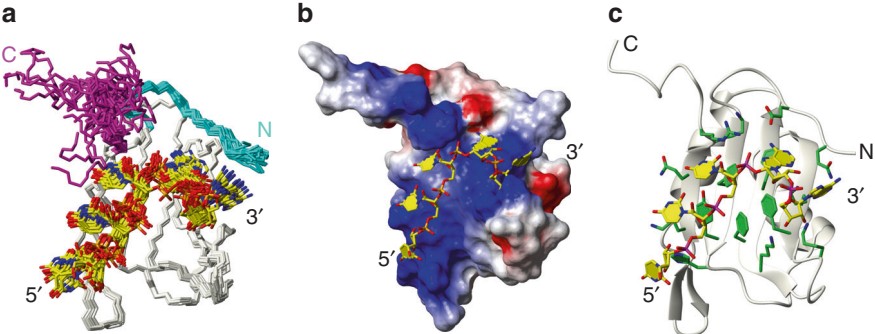

**Fig. 2** Overview of the solution structure of CUGB-BP2 RRM3 bound to 5′-UUUAA-3′ RNA. **a** Overlay of the 20 lowest energy structures of the CUG-BP2 RRM3/5′-UUUAA-3′ complex. The N- and C-termini are represented in *cyan* and *magenta*, respectively. **b** van der Waals surface representation of RRM3 with the RNA in sticks. Heavy atoms are colored according to the electrostatic potential (*blue*: positive; *red*: negative; *white*: neutral). **c** Ribbon representation of RRM3 with RNA and interacting residues of the protein in stick representation. The protein carbon bonds are colored in *green* and the RNA in *yellow*

**Dynamics of aromatics residues at the binding surface**. We have demonstrated that CUG-BP2 RRM3 adopts the UP conformation in both the free form and when bound to 5′-UUUAA-3′, whereas it is in the DOWN conformation when bound to 5′-UGUGUG-3′. This indicates that aromatic side-chains motions are critical for RNA discrimination. To gain more insight on the conformational equilibrium of aromatic side-chains at the binding surface, we measured the $\chi 1$ dihedral angle (defined by the atoms N–C$^\alpha$–C$^\beta$–C$^\gamma$) that describes the side-chains orientation relative to the protein backbone using spin-echo difference 2D $^1$H–$^{15}$N-HSQC experiments[14] and extracted $^3$J NC$^\gamma$ and $^3$J C′C$^\gamma$ scalar coupling constants of aromatic residues whose magnitude are dependent on $\chi 1$ (Fig. 4a, Supplementary Fig. 5 and Supplementary Table 2). By assuming a three-site jump model and interpreting the $^3$J NC$^\gamma$ and $^3$J C′C$^\gamma$ scalar coupling constants as reporting on the averaging between the three rotamers gauche-, gauche + and trans centered on the $\chi 1$ values + 60°, −60° or 180°, we could extract the average population of each rotameric state using the Karplus parametrization published by Tuttle et al[15]. and quantitatively characterize the conformational differences. The major rotamers determined for the aromatic residues of RRM3 wild-type free or in complex with either 5′-UUUAA-3′ or 5′-UGUGU-3′ mostly agree with the conformations observed in the corresponding structures (Fig. 4b, c). In free RRM3 wild-type, Phe426, Tyr428, His429 and Phe455 are mostly in the gauche + rotamer, whereas the trans population of Phe468 is close to 50%. Upon binding of 5′-UUUAA-3′ RNA, the picture remains unchanged except for Phe468 whose gauche-population increases. In contrast, the presence of 5′-UGUGU-3′ RNA shifts the Phe426 rotamer from gauche + to mainly trans and the fractions of the trans rotamer also increase for both Tyr428 and Phe455. It is noteworthy that, with the exception of His429 which remains in the gauche- conformation in all three samples and Phe426 and Phe468 in the RRM3/5′-UGUGU-3′ complex, all the aromatic side-chains at the binding interface are dynamic and populate more than one rotamer.

While the $^3$J coupling experiments provide evidence for dynamic rotamer averaging of the individual, aromatic residues, MD can elucidate the mechanisms underlying aromatic side-chain rearrangements. We performed simulations of the RRM3 wild-type free and of the RRM3/5′-UUUAA-3′ and RRM3/5′-UGUGUG-3′ complexes. In all cases, the aromatic side-chain mostly remain in their initial conformation but in some trajectories, the aromatic residues exhibit stochastic changes in conformation and rotate about the $\chi 1$ angle (Supplementary Table 3). Notably, Phe466, whose backbone amide is not observable due to exchange broadening in both free and bound wild-type RRM3, freely fluctuates between gauche- and trans conformations on a sub-nanosecond timescale (Supplementary Fig. 6). Most interestingly, in the last 100 ns, a simulation of the free RRM3 wild-type exhibits a complete transition from the UP conformation of the aromatic residues to the DOWN conformation (Fig. 5). A trans-to-gauche-rotation of Phe466 is followed by the change of the Phe426 $\chi 1$ angle from gauche + to trans, which triggers the transition of Tyr428 into trans. This allows the Phe468 side-chain to take on a gauche + orientation, moving into the space previously vacated by Phe426 and finally, Phe455 rotates from gauche + to trans completing the rearrangement. Furthermore, removing the RNA in the simulation of the RRM3/5′-UGUGUG-3′ complex, results in the binding interface that is initially in the DOWN state rapidly reverting to the UP state, suggesting that intermolecular contacts provided by the RNA are essential for the stabilization of the DOWN conformation.

Overall, the NMR data and the MD simulations illustrate that the RNA-binding surface can exist in multiple states, including UP and DOWN states and that the aromatic residues conformations are stabilized by intra- or intermolecular contacts. The in silico results show that the transition from the UP to the DOWN state is not a 'global' event, but rather a multiple state process where the dynamics and conformation of each aromatic residue influences the conformation of its neighbors.

**Aromatic side-chains rearrangement and affinity increase**. To assess the significance of the protein–RNA interactions identified in the structure of CUG-BP2 RRM3 in complex with 5′-UUUAA-3′ RNA, we replaced key interacting residues by alanine and determined the $K_d$ values of these variants with 5′-UUUAA-3′ RNA using NMR (Table 1, Supplementary Fig. 3). Mutating polar residues such as Lys453, Lys495, Gln497 and Lys499 had no effect or caused only minor decreases in affinity, supporting the predominant contribution of the non-specific π–π stacking to the affinity. Accordingly, the mutation of Phe426, one of the aromatics of the RNP1 motif, to an alanine led to a two-fold decrease in affinity. Surprisingly, substituting any of the non-canonical aromatics, Tyr428, His429 or Phe455 with alanine increased the binding affinity two-fold despite their observed interactions with U$_2$, U$_1$ and A$_5$, respectively, in the wild-type complex. Considering these unexpected results, we then investigated the effect of these non-canonical aromatic mutations on the binding of a high-affinity UG-rich target. We determined a $K_d$ of 1 μM for the RRM3 wild-type/5′-UGUGU-3′ complex by isothermal titration calorimetry (ITC) (Table 1, Supplementary Fig. 7), which was a more suitable method for this

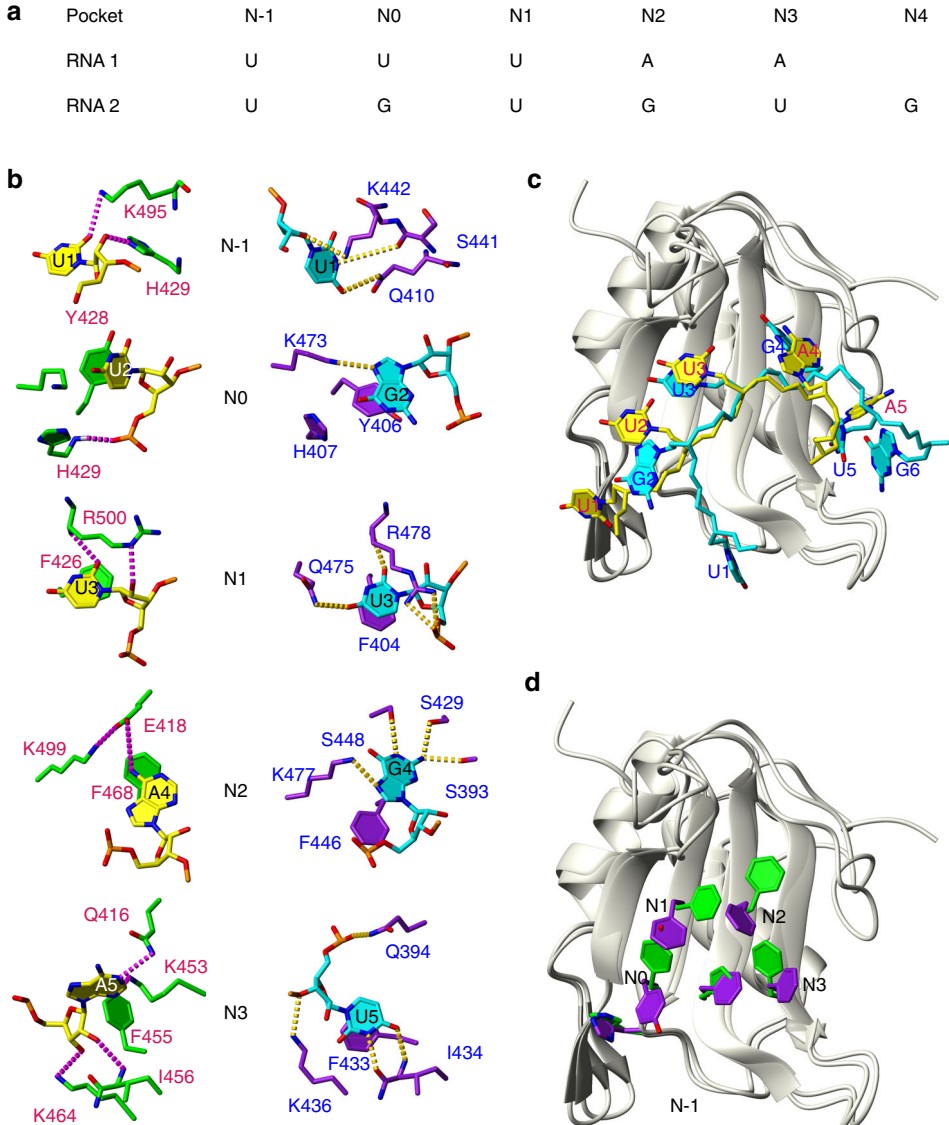

**Fig. 3** Comparison of CUG-BP2 RRM3/5′-UUUAA-3′ and CUG-BP1 RRM3/5′-UGUGUG-3′ complexes. The CUG-BP2 RRM3/5′-UUUAA-3′ complex is represented with *green* protein side-chains and the RNA in *yellow* and the CUG-BP1 RRM3/5′-UGUGUG-3′ complex with *purple* protein side-chains and the RNA in *cyan*. **a** Schematic representation of the arrangement of the nucleotides of 5′-UUUAA-3′ and 5′-UGUGUG-3′ RNAs in the binding pockets. **b** Detailed view of protein-RNA interactions in each binding pocket comparing CUG-BP2 RRM3 bound to 5′-UUUAA-3′ on the left and CUG-BP1 RRM3 bound to 5′-UGUGUG-3′ on the *right*. **c**, **d** Overlays of the complexes with protein backbone shown in *ribbon* and aromatic protein side-chains and the RNAs represented as *sticks*. For clarity, the bonds to OP1, OP2 and O4′ are removed. (**c**) shows the RNA and (**d**) shows the aromatic side-chains

higher-affinity complex. The F455A mutation resulted in a $K_d$ of 10 μM, which is a 10-fold decrease in affinity compared to the wild-type. In contrast, mutations of either Tyr428 or His429 yielded a three-fold higher affinity for the RNA than the wild-type.

Comparison of the 2D $^1$H–$^{15}$N HSQC spectra of the non-canonical aromatic mutants of RRM3 to that of the wild-type in the free state shows that the substitutions of Tyr428, His429 and Phe455 to alanine induce large chemical shift perturbations that propagate from the mutation site through the whole β-sheet (Supplementary Fig. 8). Consistent with this, the $^3$J NC$^γ$ and the $^3$J C′C$^γ$ scalar couplings of aromatic residues exposed on the β-sheet surface (e.g., Phe426, Phe468) of Y428A, H429A and F455A deviate from the wild-type values (Fig. 4a). Analyzing the rotamer populations, it appears that the mutations of non-canonical aromatics to alanine cause a conformational shift towards the DOWN state. In all the mutants, there is an increase

in the trans population of Phe426 and gauche + population of Phe468 in the free form relative to the wild-type and upon binding of 5′-UUUAA-3′, these trends are enhanced. For example, when the Y428A mutant binds the 5′-UUUAA-3′ RNA, Phe426 χ1 is almost completely in the trans rotamer state, Phe468 in the gauche + state and the trans population of Phe455 χ1 increases. In silico, mutating Tyr428, His429 or Phe455 to alanine in the free protein did not affect the dynamic behavior of the aromatic side-chains, which remained in the UP conformation on the timescale of the simulations. However, substitution of these residues to alanine in the free protein removes steric hindrances and causes the loss of van der Waals interactions between the aromatic residues. Thus, the mutations are expected to change the transition rates and equilibria between the different conformations at the binding interface. We note that the MD simulations, which were carried out for a maximum of 1 μs sample a shorter time scale than the NMR experiments

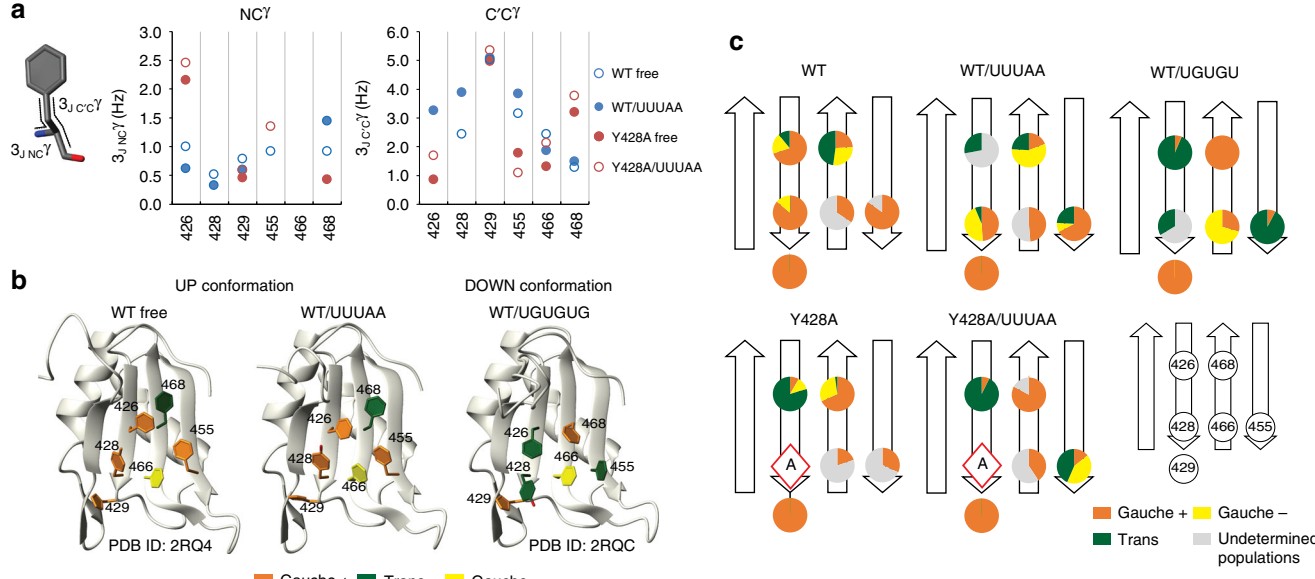

**Fig. 4** RRM3 aromatic residues scalar coupling and rotamer population. **a** $^3$J NC$^\gamma$ and C'C$^\gamma$ scalar coupling values for aromatic residues of CUG-BP2 RRM3 wild-type and Y428A mutant in the free forms and the Y428A mutant bound to 5′-UUUAA-3′. Due to the similarity in the behavior of all three mutants, only the Y428A are presented (results for the H429A and F455A mutants can be found in Supplementary Table 2 and Supplementary Fig. 7). Missing $^3$J values are not available due to exchange broadened resonances or overlap. **b** Illustration of the UP and DOWN conformations. RRM3 free (PDB ID: 2RQ4) or bound to 5′-UUUAA-3′ (this study) is in the UP conformation (Phe426 (gauche +), Tyr428(gauche +), His429(gauche +), Phe455 (gauche +), Phe466(trans), Phe468(trans)). RRM3 bound to 5′-UGUGUG-3′ (PDB ID: 2RQC) is in the DOWN conformation (Phe426(trans), Tyr428(trans), His429 (gauche +), Phe455(trans), Phe466(gauche-), Phe468(gauche +)). **c** Aromatic residue rotamer populations at the binding interface. The pie chart shows the $\chi_1$ rotamer populations determined from the $^3$J NC$^\gamma$ and $^3$J C'C$^\gamma$ scalar coupling constants for aromatic residues at the binding interface. The populations are not determined due to lack of $^3$J coupling data are colored in *grey*. Mutated positions are indicated by a *red diamond*

measuring the coupling, which report on a population weighted average of the conformations visited during the time in which the scalar couplings are active; generally, tens of milliseconds.

Alanine scanning mutagenesis reveals the key role played by the aromatic residues of CUG-BP2 RRM3 in the modulation of RNA affinity. Here, the predicted decrease in affinity reported in many previously studied RRM-RNA interactions[16] caused by the removal of a bulky non-canonical aromatic side-chain and loss of π–π stacking or hydrophobic interactions is compensated by the side-chain rearrangement induced by the mutations. The aromatic residues at the binding surface of Y428A, H429A and F455A, in both free and complexed forms, shift their rotamer populations towards those of the DOWN state. Thus, the increase in RNA-binding affinity of the non-canonical aromatic mutants compared to wild-type is correlated with a shift of the aromatic side-chains conformations at the binding interface from the UP towards the DOWN state.

**Functional implications of aromatic side-chain rearrangement**. We established that RRM3 can bind to AU-rich motifs from the 3′-UTR of COX-2 mRNA and could potentially contribute to the translation regulation of COX-2 mRNA in vivo. To validate this hypothesis, we adapted a luciferase reporter gene assay previously published[5]. The authors established that the first 60 ARE-containing nucleotides of the 3′-UTR of COX-2 mRNA are responsible for translation inhibition mediated by CUG-BP2. In our assay, we transfected HEK-293T cells with wild-type or mutated CUG-BP2, the isolated RRM12 or RRM3 and the luciferase reporter gene carrying the first sixty nucleotides of COX-2 3′-UTR (Fig. 6a, b). We observed that full-length wild-type CUG-BP2 causes a 60 % decrease in the translation of the luciferase mRNA (Fig. 6c and Supplementary Fig. 9). Mutating the canonical Phe426, which had decreased the RNA-binding affinity of

RRM3 showed little effect on the activity of the full-length protein. In contrast, mutation of the non-canonical Tyr428, which increases the affinity for 5′-UUUAA-3′ RNA in vitro leads to a greater translation inhibition in our cellular assay. These results are consistent with the impact of the mutation on the population of aromatic side-chain conformations at the binding interface shifting them towards the more tightly binding DOWN conformation. The resulting increased affinity of RRM3 for AU-rich motifs can help to explain why we observe enhanced translation inhibition in the cellular assay.

**Discussion**

We solved the solution structure of CUG-BP2 RRM3 in complex with 5′-UUUAA-3′. This structure along with our mutational studies and the CUG-BP1 RRM3/5′-UGUGUG-3′ complex solved by Tsuda et al[11] provide detailed structural information on the RRM3 RNA-binding modes for these two distinct RNA sequences. They also demonstrate the broad specificity of the domain for various uridine-rich RNAs and the ability of the RRM to adapt its binding interface to different RNA targets.

Although the nucleotides of 5′-UUUAA-3′ and 5′-UGUGUG-3′ RNAs are accommodated in the same binding pockets of the protein, the conformations of the aromatic side-chains at the binding interface differ between the two complexes. RRM3 has a lower affinity for the 5′-UUUAA-3′ RNA, which is bound in the UP conformation. In this conformation, $U_2$ can optimally stack with Tyr428 in gauche +. However, $U_3$, which stacks on Phe426, is too far away from Gln497 to form a hydrogen-bond and the stacking surface of $A_4$ on the trans rotamer of Phe468 (UP) does not have the optimal geometry. In contrast, RRM3 has significantly more affinity for the UG-rich RNA, which is reflected by the higher number of intermolecular contacts comprising four additional hydrogen-bonds and one additional stacking

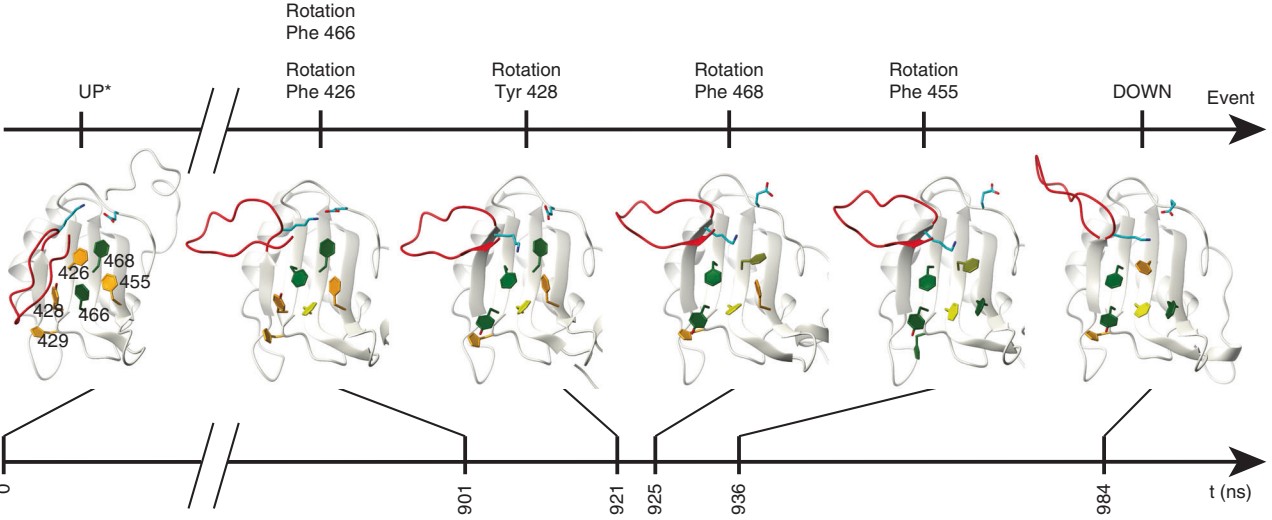

**Fig. 5** RRM3 molecular dynamics simulation. Snapshots of the free RRM3 wild-type MD simulation showing the transition from UP to DOWN conformation. The C-terminus is colored in *red*, aromatic residues in the trans, gauche + and gauche − conformation are colored in *green, orange* and *yellow,* respectively. Glu418 and Lys499 are depicted in *cyan*. Time points and key events in the side-chain rearrangement are indicated. * All residues are in the UP state except of the mobile Phe466

interaction (Supplementary Table 1). In this complex, the binding interface adopts the DOWN conformation and the Tyr428 side-chain takes on the trans conformation to stack efficiently with $G_2$. Moreover, $G_4$ has a larger stacking surface on the gauche + rotamer of Phe468 (DOWN) than that of $A_4$ observed in the RRM3/5′-UUUAA-3′ complex. Finally, $U_5$ can stack on Phe455 due to its trans rotamer orientation in the DOWN conformation. Our data demonstrate that CUG-BP2 RRM3′s discrimination between the various RNAs depends not only on the nature and the number of the intermolecular contacts formed in the complex, but also on the conformational states of the aromatic residues in the free protein and their ability to reorient to adopt the DOWN conformation, which is more favorable for binding purine containing RNA since better stacking and more hydrogen-bonds can be formed.

The NMR and MD studies presented here show that in the free form of the RRM, the binding interface exists in multiple states that are characterized by several aromatic side-chains adopting different conformations ($\chi$1 rotamers). The UP conformation, which is highly populated in the free state is stabilized by intra-molecular interactions and is bound by AU-rich RNAs with low affinity. In contrast, high-affinity targets (UG-rich RNAs) can select the more scarcely populated DOWN conformation and stabilize it by providing more favorable intermolecular interactions. This mechanism of conformational selection allows CUG-BP2 RRM3 to discriminate efficiently between different sequences rich in uracils and purines depending on whether they are able to stabilize the DOWN conformation or not. Our results are consistent with a model in which the aromatic side-chains at the RRM-binding surface exist in an ensemble of interconverting conformations, which are tunable by the binding of an RNA target. Scalar coupling data and MD simulations indicate that although the UP and DOWN states are most prominent, inter-conversion to other side-chains rotamer states takes place. In the free form, mutants replacing non-canonical aromatics by the less bulky alanine exhibit a conformational shift towards a majority of the DOWN conformation compared to the wild-type RRM3 (Fig. 7a). Consistent with the higher affinity of the DOWN conformation, titrations show that Y428A and H429A mutants can bind both 5′-UUUAA-3′ and 5′-UGUGU-3′ more tightly than the wild-type RRM despite the loss of one or more potential

interactions. F455A is an exception showing decreased affinity for 5′-UGUGU-3′. The latter complex highlights that the gain in affinity caused by the side-chain rearrangement must compensate the loss of potential stacking interactions. Furthermore, in the context of the full length CUG-BP2, the non-canonical Y428A mutation leads to a greater translation inhibition of the ARE reporter in living cells compared to wild-type, whereas the canonical F426A mutation shows no effect on the activity. Taken together, our observations illustrate the importance of the con-formational landscape of the free state of the RRM for complex formation.

A survey of the coordinates of single and tandem RRMs with canonical aromatics deposited at the Protein Data Bank shows that most of the free RRMs adopt side-chain orientations consistent with the DOWN conformation (Supplementary Table 4). However, certain RRMs such as PABPC1[17] and hnRNP G[18] show evidence of conformational averaging of the aromatic side-chains at the β-sheet surface in the free form. For example, the high-resolution structure of hnRNP A1 tandem RRM12 reveals that the canonical aromatic residues of RRM1, Phe17 and Phe52, adopt both the UP and DOWN conformations in two alternative states detected in the electron density[19]. RRM1 has also been crystallized in the DOWN conformation or in a mixed state where Phe17 takes on the orientation observed in the DOWN conformation and Phe59 the one observed in the UP con-formation. $NC^\gamma$ and $C'C^\gamma$ scalar coupling measurements and a calculation of $\chi$1 rotamer populations confirmed the dynamic nature of Phe17 side-chain in solution (Supplementary Fig. 10 & Supplementary Table 5). Furthermore, it rigidifies upon binding to 5′-UUAGGUC-3′ and adopts the orientation observed in the DOWN conformation. This is evidence that the UP to DOWN aromatic side-chain rearrangement at the β-sheet surface induced by interaction with nucleic acids is a mechanism that occurs not only in RRMs bearing non-canonical aromatics. More generally, non-aromatic side-chains can also show conformational aver-aging and may influence the equilibrium between high- and low-affinity binding similarly. Thickman et al[20]. have previously demonstrated that pre-existing alternative conformations of polar residues of U2AF65 are selectively stabilized by RNA binding.

Irrespective of their free conformation, most RRMs with canonical aromatic residues interact with RNA in the DOWN

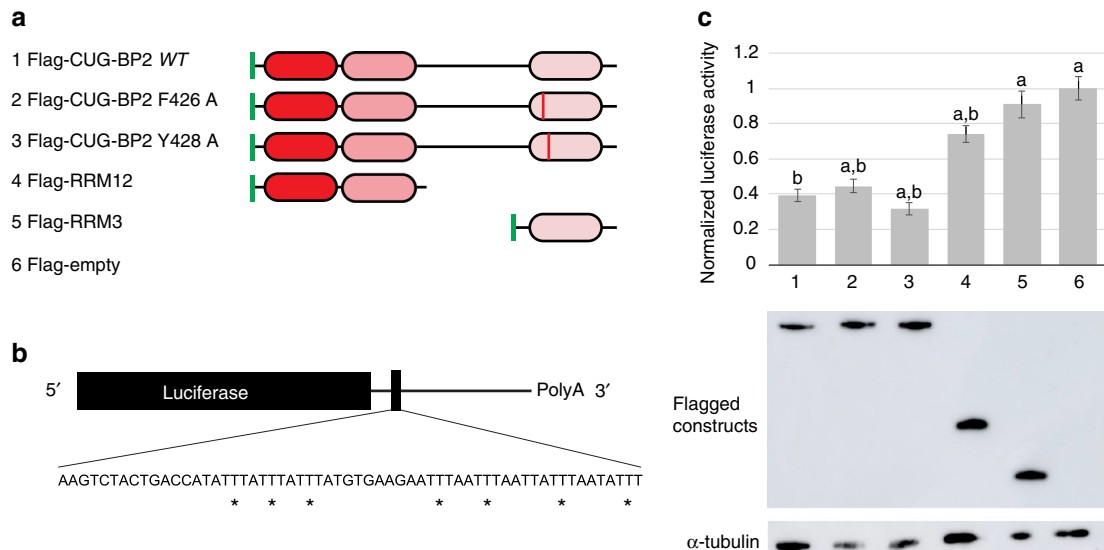

**Fig. 6** Translation inhibition of COX-2 mRNA by CUG-BP2. **a** CUG-BP2 constructs. The flag-tag is depicted as a *green bar* and the RRMs as *rounded rectangles*. **b** Luciferase reporter gene. The coding sequence of the renilla luciferase is fused to the first sixty nucleotides of COX-2 mRNA 3′-UTR. **c** Luciferase reporter assay. *Upper panel*: normalized renilla/firefly luciferase activities for the different CUG-BP2 constructs. All the values are normalized to the negative control, which is the empty vector. Statistical significance was tested using the student test (a: significantly different from WT, $p < 0.05$; b: significantly different from control, $p < 0.05$). *Error bars* represent s.d. *Lower panel*: western blot

conformation with the notable exception of CUG-BP2 RRM3 bound to 5′-UUUAA-3′. Interestingly, RRMs that show aromatic side-chain rearrangement upon RNA binding (Supplementary Table 6) share some common features. Most of them have moderate affinity (µM range) and bind RNA nucleotides repeats e.g. polyA, UG repeats and AREs. Based on our structural work and cellular assay, it seems plausible that the UP conformation might allow these RRMs to recognize regions containing repeated sequences with the multi-register binding increasing the overall affinity while retaining the ability to shift along the RNA[21] (Fig. 7b). This would agree with the register shift phenomenon and higher affinity we observed for CUG-BP2 RRM3 bound to 5′-AUUUAAUU-3′. This low sequence-specific binding mode of the UP conformation could also enable these RRMs to rapidly probe the RNA for high-affinity motifs by diffusing along a one-dimensional path (the single-stranded RNA sequence). When these motifs are encountered, a switch to the DOWN conformation would allow tight binding. This two-step mechanism of finding cognate targets involves only side-chain rearrangements within one RRM which contrasts with other RRM-containing proteins that use tandem domains for switching to a tight binding mode. For example, U2AF65 tandem RRMs populate a 'closed' state where the binding of the first RRM is precluded by an interaction with the second RRM[22]. When binding to an optimal polypyrimidine tract, the RRMs can switch to an 'open' state where both RRMs form an extended basic RNA-binding surface. Similarly, the CPEB tandem RRMs discriminate between poly (U)$_{3-4}$ stretches and cytoplasmic polyadenylation elements through a Venus fly trap mechanism[23]. Overall, intra-domain rearrangements like in CUG-BP2 RRM3 may be an additional important mechanism for RNA discrimination by RRMs.

In the present work, we established that the RNA-binding surface of several RRMs can exist in distinct states and transitions between different side-chain conformations at the binding surface is an effective way to discriminate between different RNAs and fine tune the affinity for various sequences. In addition, we show how mutations, which eliminate aromatic groups at the binding interface, can tune the conformational landscape of surface side-chains towards a tighter binding state. In the future, it will be

crucial to investigate more systematically how side-chain conformational equilibria impact RNA discrimination to further understand the underlying rules of protein-RNA interactions.

## Methods

**Cloning and purification of CUG-BP2 RRM3 and mutants.** CUG-BP2 RRM3 (residues 416–508) was cloned into pET 28a vector (Novagen) between the NdeI-XhoI restriction sites. To facilitate protein quantification, the C-terminal tyrosine was replaced by a tryptophan by Quick-change™ Site-Directed Mutagenesis. Other mutants where obtained using the same method. The primers used in this study are provided in the Supplementary Table 7.

The plasmid was transformed in BL21 DE3 Codon + *E. coli* cells (Agilent Technologies). Pre-culture and culture were carried out at 37 °C in M9 minimum media containing 1 g.L⁻¹ of ¹⁵N NH₄Cl and 8 g.L⁻¹ of unlabelled glucose or 4 g.L⁻¹ of ¹³C labelled glucose. The media was complemented with kanamycin and chloramphenicol. At an OD₆₀₀ of 0.6–0.8, the culture was cooled down to 20 °C and induced with IPTG to a final concentration of 1 mM. Cells were collected after 12–16 h at 7878 x g for 30 min at 4 °C. The protein was purified by Ni-affinity chromatography using an imidazole gradient on an ÄKTA Prime purification system (Amersham Biosciences) or step-wise by gravity flow. The protein-containing eluate was concentrated and exchanged into a low salt buffer. Removal of contaminant nucleic acids was achieved by cation exchange chromatography (HiTrap SP HP, GE Health care). After another concentration step, the protein was incubated for 1 h at room temperature with Ambion RNAse inhibitor and subsequently purified by size exclusion in NMR buffer [K₂HPO₄/KH₂PO₄ 20 mM (pH 5.8), NaCl 10 mM, β-mercaptoethanol 0.1%, DEPC treated] and concentrated to 0.6–2 mM.

Protein concentration was determined by absorbance measurement at 280 nm wavelength and application of the Lambert-Beer's law.

**Purification of hnRNPA1 RRM1.** hnRNP A1 RRM1 (residues 416–508) cloned into pET 28a vector (Novagen) was a gift from Pierre Barraud.

The plasmid was transformed in BL21 DE3 Codon + *E. coli* cells (Agilent Technologies). Pre-culture and culture were carried out at 37 °C in M9 minimum media containing 1 g.L⁻¹ of ¹⁵N NH₄Cl and 4 g.L⁻¹ of ¹³C labelled glucose. The media was complemented with kanamycin and chloramphenicol. At an OD₆₀₀ of 0.6–0.8, the culture was cooled down to 30 °C and induced with IPTG to a final concentration of 0.5 mM. Cells were collected after 6 h at 7878 x g for 30 min at 4 °C. The protein was purified by Ni-affinity chromatography using an imidazole gradient on an ÄKTA Prime purification system (Amersham Biosciences). The protein-containing eluate was concentrated and dialyzed against hnRNP A1 NMR buffer [Na₂HPO₄/NaH₂PO₄ 10 mM (pH 6.5), β-mercaptoethanol 10 mM, DEPC treated] and concentrated to 2 mM. Protein concentration was determined by absorbance measurement at 280 nm wavelength and application of the Lambert-Beer's law

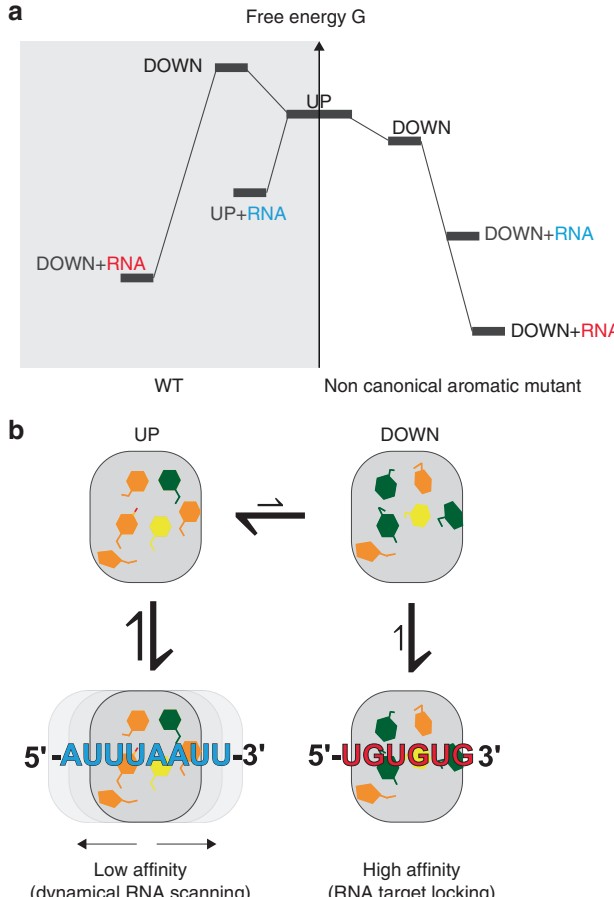

**Fig. 7** CUG-BP2 RRM3 affinity modulation model. **a** Free energy diagram of RRM3 wild-type and the aromatic mutants free and in complex with RNA. High-affinity RNA is depicted in *red* and low-affinity RNA in *blue*. **b** Four-state model of RRM3 in UP or DOWN conformations free or bound to RNA. The aromatic residues in the trans, gauche, gauche + and gauche- conformation are colored in *green, orange* and *yellow*, respectively

**Oligonucleotides preparation and complex preparation**. Oligonucleotides were purchased from Dharmacon, Inc and deprotected with the provided deprotection buffer at 60 °C for 30 min. The RNA was frozen in liquid nitrogen, lyophilized and stored at −20 °C. For NMR studies, the RNA was re-suspended in NMR buffer.

To determine the solution structure of CUG-BP2 RRM3 in complex with 5′-UUUAA-3′, the protein and the RNA were mixed at ratios of 1:1 or 1:3. At the latter ratio, 96% of the protein is saturated based on calculation of the expected fraction of free protein by using the $K_d$ value obtained from $^{15}$N-HSQC titrations.

**Cellular Assay and Western blots**. The flag-tag sequence was inserted between NheI and HindIII in the pcDNA3.1 vector. CUG-BP2 full length, RRM12 and RRM3 coding sequences were cloned into the modified pcDNA3.1 + vector between the HindIII and XhoI restriction sites. The F426A and Y428A mutants were obtained by site directed mutagenesis.

The first 60 nucleotides of the murine COX-2 3′-UTR (which has 80% sequence identity to the human COX-2 3′-UTR) were amplified from a pGL3 plasmid which was a generous gift from Aubrey Morrison[24], by using the 5′-CTAGGCGACTC GAGGATCGCCGTGTAATTCTA-3′ and 5′-TGGCCGGCGGCCGCTATCATGT CTGCTCGAAG-3′ forward and reverse primers, respectively to introduce XhoI and NotI sites. The PCR insert was cloned into the 3′-UTR of the renilla luciferase gene in the psiCHECK 2 vector (Promega).

HEK-293T (ECACC No. 85120602) cells were grown in Gibco D-MEM medium supplemented with GlutamaxTM, 10% fetal bovine serum in a humidified chamber at 37 °C with 5% $CO_2$.

For the luciferase reporter gene assay, HEK-293T cells were seeded in 24 wells plates at day0. At day1, they were transfected transiently using lipofectamine 2000 (Invitrogen) with 0.1 μg of the luciferase reporter gene and 0.3 μg of the relevant CUG-BP2 construct. After 24 h, cell lysis and luciferase activity determination were done as per the manufacturer's instructions (Dual-Luciferase Reporter Assay System; Promega) using a Berthold Mithras LB 940. The renilla activity was normalized to the luciferase activity, is presented as relative luciferase units relative

to control with ± the standard error of the mean (SEM) values. The significance of the differences between the samples was tested by a *t*-test. Assays were performed in three replicates and experiment were repeated 3 times.

Western blot analysis of the whole-cell lysate was performed by using the monoclonal anti-FLAG M2 antibody produced in mouse (Sigma) (1:2500 in TBST- 5% milk blotting powder) and monoclonal anti-α-Tubulin antibody produced in mouse (1:20000 in TBST- 5% milk blotting powder). After secondary antibody incubation (Anti-Mouse IgG (whole molecule) Peroxidase antibody produced in rabbit, in 1:50,000 in TBST- 5% milk blotting powder), western was detected by using the Amersham ECL Prime Western Blotting Detection Reagent (GE Healthcare). Western blots of two biological replicates were done.

**NMR spectroscopy**. NMR experiments were performed on AVANCE III (600, 700,750, 900 MHz) and AVANCE III HD 600 MHz Bruker spectrometers. For NMR measurements with sample volumes below 450 μl 5 mm NMR tubes from Shigemi Inc. were used (Allison Park, USA). For larger samples, 5 mm NMR tubes from ARMAR AG (Döttingen, Switzerland) were used. Typical NMR protein concentrations were 0.6−2 mM in NMR buffer and 10% $D_2O$ (v/v). The data were processed using Topspin 2.1 and 3.0 (Bruker) and analysed using SPARKY 3.113[25].

The backbone resonance assignments of free CUG-BP2 RRM3 were previously determined by Fred Damberger and Neel Bavesh. The backbone and side-chain assignments CUG-BP2 RRM3 in complex with 5′-AUUUAAUU-3′ were obtained with the following experiments: 2D$^{15}$N–$^1$H HSQC, 2D $^{13}$C–$^1$H HSQC, 3D HNCA, 3D CBCACONH, 3D 3D (H)C(CCO)NH TOCSY, 3D H(CCCO)NH TOCSY, 3D NOESY $^{15}$N–$^1$H HSQC and 3D NOESY $^{13}$C–$^1$H HSQC, all recorded in $H_2O$ at 40 °C. To sequentially assign 5′-AUUUAAUU-3′ unlabeled RNA bound to CUG-BP2 RRM3, we recorded the following experiments in $D_2O$ at 40 °: 2D $^1$H–$^1$H TOCSY, 2D F1 $^{13}$C-filtered, F2 $^{13}$C-filtered $^1$H–$^1$H NOESY and natural abundance 2D $^{13}$C–$^1$H HSQC. Intermolecular NOEs of the CUG-BP2 RRM3-RNA complex were obtained from a 2D $^1$H–$^1$H NOESY, 2D F2 $^{13}$C-filtered $^1$H–$^1$H NOESY and 3D F1 $^{13}$C-filtered $^1$H–$^1$H NOESY, all recorded in $D_2O$.

For the CUG-BP2 RRM3 in complex with 5′-UUUAA-3′, we recorded a 3D NOESY $^{15}$N–$^1$H HSQC and two 3D NOESY $^{13}$C–$^1$H HSQCs centered on aliphatic or aromatic $^{13}$C signals, all recorded in $H_2O$ at 25 °C. Protein backbone and side-chain assignments were derived from the assignments obtained for the complex with the longer RNA and adapted to the NOESY spectra recorded on CUG-BP2 RRM3 in complex with 5′-UUUAA-3′. To determine the protonation state of the histidines, we recorded long-range 2D $^{15}$N–$^1$H HSQCs[26]. We measured 3D HNHA experiments to quantify the $^3$JHN–HA coupling constant and determined ψ dihedral angles[27, 28]. To sequentially assign 5′-UUUAA-3′ RNA and obtain intermolecular NOEs, we recorded the following experiments in $D_2O$ at 25°: 2D $^1$H–$^1$H TOCSY, 2D F2 $^{13}$C-filtered $^1$H–$^1$H NOESY and natural abundance 2D $^{13}$C–$^1$H HSQC. Intermolecular NOEs were obtained by using a 2D $^1$H–$^1$H NOESY, 2D F2 $^{13}$C-filtered $^1$H–$^1$H NOESY and 3D F1 $^{13}$C-filtered $^1$H–$^1$H NOESY $^{13}$C–$^1$H HSQC.

Measurements of the aromatic $^3$J NC$^\gamma$ and $^3$J C′C$^\gamma$ coupling constants was carried out by recording spin-echo difference constant time $^{15}$N–$^1$H HSQC and HN(CO)CG experiments[29], respectively. The pulse sequences were modified compared to the published version. The 180° $^{13}$C shaped pulses were changed from G3[30] to Q3[31] to improve selectivity. The 90° C′ pulses were changed from rectangular to Q5[31] shape. At 600 MHz, the durations of the Q3 C′ and C$^\gamma$ 180° pulses, of the Q3 C$_{aliphatic}$ 180° pulses and of the Q5 C′ 90° pulses are 760 μs, 512 μs and 760 μs, respectively. The reference and the coupled experiments were interleaved and all the measurements were repeated at least three times.

The coupling constants were calculated from the relationship $I_b/I_a = \cos$ $(2\pi JC^\gamma\tau)$, where $I_a$ is intensity of an H–N correlation in the reference spectrum and $I_b$ is the intensity of the same peak in the spectrum where the relevant coupling was active for either the amide signal of the i + 1 residue in the $^3$J C′C$^\gamma$ experiment or for the i residue in the $^3$J NC$^\gamma$ experiment, where i is the residue whose coupling is determined. In $^3$J C′C$^\gamma$ experiments, τ was set to 40 or 50 ms for CUG-BP2 RRM3 and hnRNP A1 RRM1, respectively. In $^3$J NC$^\gamma$ experiments τ was set to 65 ms. The values of τ used were determined empirically to maximize the intensity for signals of interest when calculating the difference between reference and coupled spectrum.

**Structure calculation**. In NOESY spectra, NOEs originating from the aromatics at the binding interface were manually assigned to derive upper limit restraints. Those were input for the iterative process of automated peak picking of the 3D NOESY $^{15}$N–$^1$H HSQC, and the 3D NOESY $^{13}$C–$^1$H HSQCs performed by ATNOSCANDID[32–34]. The peak lists generated by ATNOSCANDID but not the manually assigned aromatic NOEs were carried over to the next step. Automated NOE assignment of intra-protein NOEs was achieved using the macro NOEASSIGN within CYANA 3.0[35] and manual curation.

RNA intramolecular and intermolecular NOEs were manually assigned and calibrated using a 1/d[6] relationship. Structure calculation was carried out using CYANA 3.0 and included distance restraints, TALOS[36] derived dihedral angle constraints in agreement with the determined $^3$J HN-HA coupling constants, intra-protein hydrogen-bond constraints derived from hydrogen–deuterium exchange experiments on the amide protons and RNA dihedral angle constraints based on $^1$H1′–$^1$H2′ coupling efficiency in homonuclear 2D TOCSYs measured with short

TOCSY mixing times. 200 structures were calculated with 20000 torsion angle dynamics steps per conformer. The best 50 conformers were subsequently refined in implicit water using the force field rna.ff99SB in the SANDER module of AMBER 12[37, 38]. Finally, the 20 conformers with the lowest amber energy were submitted to CING[39] for structure validation.

**Dissociation constant determination**. ITC experiments. ITC was performed using VP-ITC calorimeter (MicroCal LLC). Protein was injected into the microcalorimeter cell containing the RNA. The protein and RNA samples were dialysed against the same buffer [KPi 20 mM (pH 5.8), NaCl 10 mM, β-mercaptoethanol 0.1%, DEPC treated]. Wild-type and mutant RRM3 proteins (0.13–0.47 mM) were titrated by injection of 6 μl into 2 ml of 8 or 9 μM RNA. The experiments were carried out at 25 °C. The raw data were analysed using Microcal Origin 7.0 software with the single binding site model.

NMR titration. Chemical shift changes in $^1H-^{15}N$ HSQC spectra were quantified using the following equation:

$$\Delta CS_{obs} = \sqrt{\left((\delta HN)^2 + \left(\frac{\delta N}{6.41}\right)^2\right)}$$

Where $\delta HN$ and $\delta N$ are the proton and nitrogen chemical-shift differences of the same signal.

Subsequently, the NMR titration data were fitted using the following equation in Matlab

$$\Delta CS_{obs} = \Delta CS_{max}\left(\frac{(K_d + [L]_0 + c[P]_0) - \sqrt{(K_d + [L]_0 + c[P]_0)^2 - 4[L]_0 c[P]_0}}{2c[P]_0}\right)$$

where $[P]_0$ and $[L]_0$ are the concentrations of total protein and RNA, respectively, $K_d$ the dissociation constant, $\Delta CS_{max}$ the maximum frequency difference, $\Delta CS_{obs}$ the observed frequency at a given titration point and c a correction coefficient for the protein concentration determined by absorbance measurements with a calculated extinction coefficient.

**χ1 rotamers population determination**. The $\chi 1$ rotamer populations were determined under the assumption that the values of the $^3J$-couplings reflect a population mixture of the three staggered states using the Karplus parametrization published previously[15]. Accordingly:

$$^3J_{calc,NC'} = p_{180}{}^3J_{t,NC'} + p_{-60}{}^3J_{g+,NC'} + p_{+60}{}^3J_{g-,NC'}$$

$$^3J_{calc,C'C'} = p_{180}{}^3J_{t,C'C'} + p_{-60}{}^3J_{g+,C'C'} + p_{+60}{}^3J_{g-,C'C'}$$

$$1 = p_{180} + p_{-60} + p_{+60}$$

where $^3J_{calc}$, NC$^\gamma$ and $^3J_{calc}$, C'C$^\gamma$ are the calculated coupling constants, $p_{180}$ $p_{-60}$ and $p_{+60}$ are the populations of the respective $\chi 1$ rotamer states trans, gauche + and gauche- respectively; and $J_t$, $J_{g+}$, and $J_{g-}$ for NC$^\gamma$ and C'C$^\gamma$ are the expected coupling values for the fully populated staggered rotamer states with $\chi 1$ equal to 180°, −60°, and + 60°, respectively. Populations were fitted in Mathematica by minimization of the squared difference between the experimental and calculated $^3J$ NC$^\gamma$ and/or $^3J$ C'C$^\gamma$ coupling constants:

$$\chi^2 = \Sigma\left(\frac{^3J_{calc} - {}^3J_{exp}}{\sigma_{J_{exp}}}\right)^2,$$

where $\sigma_{J_{exp}}$ is the standard deviation based on three or more $^3J$ measurements

**Molecular dynamics simulation**. *Structure building and force field selection*. We have used the first conformers of the NMR ensembles of the free CUG-BP1 RRM3 domain structure (PDB: 2rq4)[11], its complex with (UG)$_3$ RNA (PDB: 2rqc)[11] and 5'-UUUAA-3' RNA (this work), respectively, as the starting structures for our MD simulations. The coordinate and topology files were created using the tleap module of Amber 14[40]. We have used the ff99bsc0$\chi_{OL3}$[41–44] and ff12SB[41, 45, 46] force fields to describe the RNA and the protein, respectively.

*System solvation*. The biomolecules were solvated in an octahedral box of SPC/E waters[47] with a minimal distance of 10 Å between the solute and the edge of the box. The systems were neutralized by addition of KCl salt[48], achieving a 150 mM excess-salt concentration.

*Simulation protocol*. Prior to the production simulations, the systems were minimized and equilibrated using a standard equilibration protocol for protein/ RNA systems[49]. We have then used the initial portions of the production simulations to stabilize the structure using the experimental NMR restraints[50]. This was followed by free unrestrained simulations.

The particle mesh Ewald was used for calculation of the electrostatic interactions[51, 52]. The cut-off distance for Lennard-Jones interactions was 9 Å. We have used the SHAKE algorithm[53] to constrain the covalent bonds involving hydrogen along with the HMR scheme[54], allowing a 4 fs integration step to be used. Berendsen thermostat and barostat[32] were used to maintain the systems at temperature and pressure of 300 K and 1 bar, respectively.

*Simulation analyses*. We have used the cpptraj module of Amber14[40] to analyze the simulation trajectories. The VMD program was used for visualization[55]. To evaluate the simulation agreement with the experimental data, we have calculated $(r^{-6})^{(-1/6)}$ weighted averages of the NOE distances in the simulation ensembles. These values were then compared with the experimentally measured upper bound distances of the individual NOEs.

A list of the simulation can be found in Supplementary Table 8

**Data availability**. Coordinates of CUG-BP2 RRM3 in complex with 5'-UUUAA-3' have been deposited in the Protein Data Bank under the accession code 5M8I. The corresponding chemical shift assignments and scalar couplings have been deposited in the BioMagResBank (BMRB) under the accession number 34057. The scalar couplings of CUG-BP2 RRM3 wild-type free or in complex with 5'-UGUGU-3', of CUG-BP2 RRM3 Y428A free or in complex with 5'-UUUAA-3', of CUG-BP2 RRM3 H429A free or in complex with 5'-UUUAA-3', and of CUG-BP2 RRM3 F455A free or in complex with 5'-UUUAA-3' have been deposited under the BMRB accession numbers, 27140, 27154, 27142, 27155, 27152, 27156, 27153 and 27157, respectively. The scalar coupling constants of hnRNPA1 RRM1 free or in complex with 5'-UUAGGCU-3' have been deposited under the BMRB accession numbers 27163 and 27164, respectively. Other data supporting the findings of this manuscript are available from the corresponding author upon reasonable request.

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

## Acknowledgements

This work has been supported by the Swiss National Science Foundation grant Nr. 3100A3-170130, the Ministry of Education, Youth and Sports of the Czech Republic, grant: [LO1305] and the Czech Science Foundation; grant: [P305/12/G034].

## Author contributions

F.H.-T.A. Designed the project. F.F.D. Adapted the pulse sequences for QJ coupling experiments and helped in their recording and analysis. N.d.d.K. Designed the project, cloned the protein constructs for biochemistry and cellular biology experiments, prepared the protein and RNA sample for structural studies, recorded the NMR experiments and analyzed the data, did the ITC measurements, performed luciferase assays and analyzed MD simulations. O.D. Designed the project. M.K. Designed and analyzed the MD simulations. N.R. Cloned luciferase reporter gene into psi-CHECK 2 vector, performed luciferase assays, performed and analyzed western blotting. J.S. Designed the MD simulations. All authors contributed to writing and proofreading of the manuscript.

## Additional information

**Competing interests:** The authors declare no competing financial interests.

