## [Peer Review file · Nature Communications]

Reviewers' comments:

Reviewer #1 (Remarks to the Author):

This is an interesting study of protein-RNA interactions, in which the ability of an RNA recognition motif (RRM) to bind different RNA sequences with either a low or a high affinity is explained using structural information from NMR and molecular dynamics together with ITC affinity data for the wild type protein as well as for variants where key amino acids have been mutated to alanines.

The protein uses the same binding site for different RNA sequences, but achieves the low/high affinity capability through a re-orientation of aromatic side chains in the binding site; this alters the hydrogen bonding and stacking interactions with the target RNAs. This switching mechanism is likely to be a common feature in RRM proteins, as indicated by the survey of structures in the PDB where the authors find evidence for both types of side chain conformation in other RRMs.

The study is carefully conducted using state-of-the-art methods, and the conclusions are supported by the data.

Reviewer #2 (Remarks to the Author):

The authors report a solution NMR structure of CUG-BP2 RRM3 in complex with a low affinity single-stranded UUUAA RNA ligand that is derived from an AU-rich element (AREs) in the 3' UTRs of COX-2 mRNA, the stability of which is regulated by CUG-BP2. The RNA recognition mode of RRM3 with UUUAA RNA is compared with the previously reported structure of a high affinity UGUGU RNA ligand. The authors propose a mechanism of RNA discrimination that involves distinct side chain conformations of aromatic residues (UP/DOWN) that are present in some types of RRMs. Based on mutational analysis of these aromatic residues, and a combination of affinity measurements and MD simulations the authors propose that distinct side chain conformations of aromatic residues reflect more dynamic binding with low affinity RNA ligands such as UUUAA and high affinity ligands, such as UGUGU.

Throughout the paper, the authors compare their structure with that of a nearly identical homologue with the high affinity UGUGUG ligand (Tsuda et al NAR 2009). This questions somewhat the novelty of the structural data. An interesting feature is the identification of different RNA recognition modes with low and high affinity RNA ligands involving distinct side chain arrangements in the binding surface. This aspect has already been noted by Tsuda et al. based on a comparison of distinct UP/DOWN conformations of aromatic side chains in the free and RNA bound RRM structures. Going beyond the previous work is the proposal that there UP/DOWN orientation is dynamics in the free RRM domain, although for this only limited experimental evidence is provided.

The manuscript appears technically sound, although some more experimental and analysis details should be provided. It addresses an interesting aspect of RNA recognition by RRMs, which seems to be present in few RRMs that are discussed by the authors. The differences in affinities and effects of mutations studied are very small (2-3 fold). The overall biological significance of the findings is not clear, especially considering that there are two additional RRMs present in CUG-BP, which will greatly contribute if not determine overall RNA binding affinities. No functional data are provided in this respect. Thus the manuscript seems interesting but addresses a very specific feature of RRM-RNA interactions, which might be better suited for a more specialized journal, unless additional experimental data and hints on potential functional significance are provided.

Specific comments:

- The authors compare RNA ligands with different length and sequence. The structural data involve UUUAA (current work) and UGUGUG (Tsuda et al) – can the authors rule out that there are structural differences of RNA recognition related to this?
- Table 1 compares affinities for RNA sequences, but different methods are used (NMR and ITC for

UUAAA and UGUGU ligands). The Kds listed are thus not directly comparable. NMR titrations are affected by systematic errors in case the binding regime is not purely two-state and dynamic approaching intermediate timescale – which is the case in the present study. In any case, the methods used should be clearly indicated in Table 1.

- The authors provide some hints on the presence of UP/DOWN conformations and dynamic averaging. The MD simulations are state-of-the-art, however, the dynamic averaging is likely in the us to ms timescale, thus the simulations can only hint on the conformational switch. Given that the principal novelty of the manuscript is the proposed dynamic side chain switch linked to RNA recognition, more direct experimental evidence should be provided. There are various types of NMR experiments that can more directly provide such information, such as amide and methyl autorelaxation and relaxation dispersion experiments, and experiments that specifically monitor aromatic side chain dynamics (using site-specific ¹³C labeled aromatic residues). Also, the authors do not mention whether there is evidence in NMR spectra indicated by differential line-broadening, which is expected for dynamics involving aromatic residues.
- Out of the three non-canonical aromatic residues mutated to Alanine, only F455A leads to a decrease in the affinity for 5'-UGUGU-3' while there is an increase in affinity for 5'-UUUAA-3'. Contrastingly, Y428A and H429A both lead to an increase in affinity for low and high affinity ligands. These seems to contradict the general conclusions made and the authors should comment on this.
- The authors mention in the discussion that comparison of several structures of single and tandem RRM show that majority of them are in the DOWN conformation. They should present some statistics on this.
- Aromatic residues are present in a small subfamily of RRM, so the findings seem to be representative only for a few examples. The authors mention that this mechanism of RNA discrimination might be applicable to other proteins, but they should discuss in more details at least with the other members of the same family? For example. ELAV-like (HU) proteins have only one non-canonical aromatic residue and the applicability of this mechanism using just one residue might be farfetched.
- Kielkopf and coworkers have reported alternate side chain conformations on the surface of an RRM (Thickman et al JMB 2007) – the authors should discuss their findings in the context of this.
- The analysis of aromatic side chain rotamers needs to be better documented. The authors mention that the NMR pulse sequence has been adapted from the original experiments of Bax and coworkers – if so how and why? Have the authors considered systematic errors in the determination of J-couplings due to differential relaxation/line-broadening, especially of the dynamic aromatic residues that are considered? More importantly, the rotamer analysis depends critically on the Karplus curves used – which curve has been used?
- The rotamer population derived from the 3J coupling analysis is quite different compared to the MD simulations, which mostly indicate one major rotamer – this should be discussed and clarified. A direct comparison should be provided. However, given the time scales involved it is not expected that the relatively short MD simulations will provide a faithful representation of the dynamics.
- The authors propose that the RNA bound state remains somewhat dynamic – what is the evidence for this? If so, how can the authors rule out that this dynamic features affect the structural calculation employed?
- Figures are somewhat complicated and would benefit from improving clarity.
- The authors mention that the C-terminus transiently contacts the aromatic interface and might contribute to preference for gauche+ conformation in case of F426 (due to steric hindrance). Also, its deletion in MD simulation, alters the UP/DOWN conformation transition rates. It would be interesting to see if this can be corroborated by recording NMR scalar coupling experiments on this deletion mutant.
- It is stated at 1 protein: 3 RNA ratio, 96% protein population is in bound form. This is probably extracted from the NMR titration, but it should maybe mentioned in the Methods section.
- The NMR structural statistics reported in Table S1 under energy statistics (distance restraint violation and AMBER violation energy) seems not excellent. Distance violations seem rather high? Maybe somehow the numbers reported are the absolute no. of restraints violated and not mean ± S.D.?

- Figure legends of S3/4/5 are all mixed up. This should be corrected.
- The numbering of residues is wrong in the text when compared with the Table1 (for example, residue K395A in Table is mentioned as K495 in text!).
- In the Results section, the authors mention that they have 2325 intra-protein distance restraints while in the Table S1 it says 2347.

Reviewer #3 (Remarks to the Author):

When composing your report, the following questions might assist you in writing an incisive, well-justified review.

What are the major claims of the paper?

The structure determined here describes an interaction of the RRM3 domain of CUG-BP2 with a short RNA motif. This complex is similar to the previously determined complex of CUG-BP1 bound to a high affinity RNA motif. The different interactions observed in each complex are used to explain the different affinities measured for these two RNAs,

Mutations in the RRM domain generate mutated domains with better affinity than that of wild type. To explain this observation, the authors performed many analyses to clarify the role of the relative orientation of the residues –directly or indirectly- affected by these mutations.

One concern is if the increase of affinity is due to a change in the binding mode (different nucleotides are recognized when the protein is mutated) and not only due to changes in the populated rotamers and most importantly, which is the biological relevance of studying the impact of these mutations in the affinity of the RNA recognition. Are some of these mutations observed in patients?

One concern is that the authors described that the side chains orientations at the binding interface resemble that of the unbound protein. Since the affinity is low perhaps the bound rotamers are barely detectable. Is this a possibility?

Another concern is that the NOEs were automated assigned for the protein. Were the aromatic residues manually assigned? Were some of these Chemical shifts overlapped and ambiguously assigned? Perhaps the authors can provide additional information on how all these interactions and the rotamers of the residues were determined.

Are they novel and will they be of interest to others in the community and the wider field? If the conclusions are not original, it would be helpful if you could provide relevant references. This is a common feature in the field of structural biology. The authors have made a deep study to characterize one example, that of the RRM domains.

Is the work convincing, and if not, what further evidence would be required to strengthen the conclusions?

Including the structures characterized in this work (the pdb files, the restraints used to determine them and the measurements that define the rotamers described in the paper) would contribute to a better understanding of the described results.

It will also explain the ensemble of conformations present in these structures (or the relative population of the conformers observed experimentally in the wild type). It is a bit contradictory that the presence of multiple rotamers explains the selectivity of the protein and at the same time, the interactions –even for a low affinity binding site-, are well-defined, allowing the measurement of hydrogen bonds with the RNA. Were some of these distances included as restraints during the calculation?

On a more subjective note, do you feel that the paper will influence thinking in the field? Perhaps this influence might be very specific to the field of RRM domain interactions.

Please feel free to raise any further questions and concerns about the paper.

Comments:

Line 80, high degree of homology. It should say high degree of similarity (or identity).

In several instances in the text, configuration is used instead of conformation.

ITC values need error boundaries.

We would also be grateful if you could comment on the appropriateness and validity of any statistical analysis, as well the ability of a researcher to reproduce the work, given the level of detail provided.

No need it here.

Reviewer #1 (Remarks to the Author):

We thank the first reviewer for the positive appraisal of the methodology and significance of our work.

Reviewer #2 (Remarks to the Author):

General comments

Point 1: Novelty

Tsuda et al. observed that CUG-BP1 RRM3 binding surface changes from the UP conformation in the free form to the DOWN conformation upon binding of 5'-UGUGUG-3'. Our study establishes that the UP conformation is compatible with RNA binding and to our knowledge; the RRM3/5'-UUUAA-3' complex is the first example of an RRM binding in the UP conformation. Whereas Tsuda et al. focused on binding of CUG-BP1 to the poly UG motif and suggested that RRM3 may have a similar binding mode for CUG and UA repeats, we show that it can bind an ARE motif using a novel switching mechanism at the binding interface.

In addition, we make the link between the low or high affinity binding to RNA and the UP or DOWN conformation, respectively. We also discuss the fact that the altered stacking geometry in the two modes allows for distinct hydrogen bond networks which in turn explains how different bases can be recognized in the same pocket (see pocket 1). The dynamic aspect of the free state as well the bound state is established by the averaging of χ_1 scalar couplings, which we regard as an appropriate parameter to describe a conformational change characterized by the reorientation of aromatic side-chains. We agree with this reviewer that this dynamic property of the free and bound states is intriguing new information, however it is not the only new information provided nor the most important (see above). We have revised the introduction and the discussion to highlight more clearly the novel aspects of our study.

Point 2: Biological relevance

The reviewer raised the question of the biological relevance of our manuscript. To address this, we have added new results. We performed a luciferase assay with a reporter gene carrying the ARE element contained in the first 60 nucleotides of COX-2 mRNA 3'UTR and different constructs of CUG-BP2. We demonstrate the involvement of RRM3 in translation inhibition mediated by the ARE in COX-2 3'UTR. Our results also show that mutation of non-canonical aromatics to Ala which have an increased affinity also have an increased activity *in vivo* indicating that the changes in conformational state at the binding surface of the domain may also be important in the cellular context (new fig. 6).

Specific comments

Point 1: *"The structural data involve UUUAA (current work) and UGUGUG (Tsuda et al) – can the authors rule out that there are structural differences of RNA recognition related to this?"*

Although 5'-UGUGUG-3' is one nucleotide longer than 5'-UUUAA-3', 5'-UGUGU-3' is the core sequence that is recognized by RRM3 and it occupies the same pockets as for our UUUAA pentamer. In the NMR structure of the RRM3 in complex with 5'-UGUGUG-3', the sixth nucleotide does not make any specific contact to the RRM with only one hydrogen bond mediated by the phosphate backbone. Moreover, it is poorly ordered due to the small number of constraints. For all these reasons, we are confident that the difference in length between the two RNAs does not impact our conclusions.

Point 2: *“different methods are used (NMR and ITC for UUUAA and UGUGU ligands). The K_d listed are thus not directly comparable. NMR titrations are affected by systematic errors in case the binding regime is not purely two-state and dynamic approaching intermediate timescale”*

We are aware of the points raised by the reviewer and that is why we do not directly compare values from the two different datasets (NMR & ITC) as mentioned in the legend. We observed no significant linewidth broadening for the peaks used for the K_d determination by NMR and the complexes were all in fast exchange. Thus, we are confident that the dissociation constants we measured by NMR are exempted from the systematic error mentioned by the reviewer. To confirm that the UG-rich sequence has a higher affinity than the AU-rich RNA, we also measured the dissociation constant of the RRM3/5'-UUUAA-3' complex by ITC. The K_d value was 18 μM, which is clearly larger than the value of 1 μM for RRM3/5'-UGUGU-3' complex. We now provide in the supplementary material the ITC and ¹⁵N HSQC titrations binding isotherms that were used to determine the K_d values (Figure S3 and S5).

Point 3: *“Given that the principal novelty of the manuscript is the proposed dynamic side chain switch linked to RNA recognition, more direct experimental evidence should be provided. There are various types of NMR experiments that can more directly provide such information”*

In this work, we used the population averaging of the aromatic side-chains conformation as a metric for dynamics. The approach we opted for has already been reported in the literature (Tuttle, L. M.; Dyson, H. J.; Wright, P. E. *Biochemistry* 2013, 52, 3464) and scalar couplings are often used to support the presence of dynamics and to detect averaging. Since the primary hallmark of the conformational switch is the reorientation of aromatic side-chains, we regard these measurements as an effective tool to describe it. We acknowledge that the relaxation experiments suggested by the reviewer would be interesting to carry out but amide and methyl auto-relaxation would only provide indirect evidence for the reorientation of aromatic side-chains. Experiments that specifically monitor relaxation dispersion of ¹³C aromatics constitute an interesting follow-up study, which could be used to investigate timescales of the motions, and the degree to which these timescales correlate for different aromatic residues. However, site-specific labeling of the protein and its mutants and the acquisition of the NMR data as well as their analysis would considerably delay the present publication. In addition, we feel that inclusion of this data would overload an already complex manuscript. We therefore prefer to defer such studies to a follow-up paper.

Point 4: *“Out of the three non-canonical aromatic residues mutated to Alanine, only F455A leads to a decrease in the affinity for 5'-UGUGU-3' while there is an increase in affinity for 5'-UUUAA-3'. Contrastingly, Y428A and H429A both lead to an increase in affinity for low and high affinity ligands. These seems to contradict the general conclusions made and the authors should comment on this.”*

On the one hand, substitutions of the non-canonical aromatic residues to alanines result in the loss of stacking interactions, which is energetically unfavorable. On the other hand, we observe that all three non-canonical aromatic mutants have a rearranged binding interface compared to the WT RRM, which is correlated with higher affinities. Thus, there is a balance between the gain in affinity caused by the side-chain rearrangement and the loss of stacking interactions. All the mutants have a higher affinity for 5'-UUUAA-3' but only Y428A and H429A also bind 5'-UGUGU-3' more tightly. In the case of F455A in complex with 5'-UGUGU-3', it seems that the unfavorable loss of stacking interaction is more important

than the gain obtained by favoring the DOWN arrangement. We now place more emphasis on this interpretation in the results section.

Point 5: *“The authors mention in the discussion that comparison of several structures of single and tandem RRM show that majority of them are in the DOWN conformation. They should present some statistics on this.”*

We now provide statistics regarding the RRM structural survey in the supplementary material section (Table S5).

Point 6: *“Aromatic residues are present in a small subfamily of RRMs, so the findings seem to be representative only for a few examples. The authors mention that this mechanism of RNA discrimination might be applicable to other proteins, but they should discuss in more details at least with the other members of the same family?”*

In the present paper, we define the UP and the DOWN conformations as distinct conformations of the side-chain of aromatic residues at the binding interface including canonical and non-canonical aromatics. Therefore, this definition holds true for any RRM exposing aromatic residues on the β -sheet surface.

Many RRMs display multiple aromatic side-chain conformation in the free form or side-chain reorientation upon RNA binding (see table S6). Some of them are mentioned in the paper. It is plausible that in the absence of non-canonical aromatics, other types of bulky side-chains control the conformational landscape at the binding interface. For example, in the case of HnRNPA1 (Vitali, J.; Ding, J.; Jiang, J.; Zhang, Y.; Krainer, A. R.; Xu, R.-M. *Nucleic Acids Res* 2002, 30, 1531), Vitali et al. discuss the contribution of non-aromatic residues to the rearrangement of aromatic side-chains of RRM1 RNPs. To confirm our hypothesis, we now include additional data on this protein. We have measured the NC^{γ} and $C^{\gamma}C^{\gamma} {}^3J$ couplings of the aromatic residues of hnRNP A1 and determined their rotamer populations. We observed that in RNP2, Phe17, which displays multiple side-chain conformations, exhibits dynamic averaging in the free form and is dominantly present in a single rotamer state upon RNA binding (Figure S9). Similarly to CUG-BP2 RRM3 in complex with 5'-UGUGU-3', the interaction with the RNA induces a transition from the UP to the DOWN conformation for this residue. In addition, more generally, not only aromatics can rearrange to interact with RNA. The paper suggested by the reviewer #2 illustrates this very well. To address the general implications of our results, we included additional text in the discussion.

Point 7: *“Kielkopf and coworkers have reported alternate side chain conformations on the surface of an RRM (Thickman et al JMB 2007) – the authors should discuss their findings in the context of this.”*

We have added discussion placing our work in the context of the previous work of Kielkopf and coworkers as addressed in the previous section.

Point 8: *“The analysis of aromatic side chain rotamers needs to be better documented. The authors mention that the NMR pulse sequence has been adapted from the original experiments of Bax and coworkers – if so how and why? Have the authors considered systematic errors in the determination of J-couplings due to differential relaxation/line-broadening, especially of the dynamic aromatic residues that are considered? More importantly, the rotamer analysis depends critically on the Karplus curves used – which curve has been used?”*

The 180° ¹³C shaped pulses were changed from G3.1000 to Q3.1000 to improve selectivity – we found this reduced small artifacts observed in the difference experiment at very low coupling times. 90° C' pulses were changed from rectangular to Q5.1000 shape. We have added this detail to the methods section in the supplementary material. To analyze the scalar coupling data, we used the Karplus parameters from the Tuttle et al. publication. It was previously specified in the material and methods but we now introduce it in the results section. We agree that the population values we obtained are dependent on the Karplus curve used. We have tested other Karplus parameter sets but the largest coupling values we measured were significantly larger than maximal couplings of those parameter sets, which may be because the datasets used to obtain those parameters included significant averaging effects (see Tuttle et al. 2013). Nonetheless, qualitatively they yielded distributions similar to those we describe in the paper. The differential relaxation of in-phase and antiphase magnetization terms due to spin-flips of the C^γ are expected to result in an underestimation of the actual coupling constants of less than 4% for ³J NC^γ and less than 3% for ³J C'C^γ (Loehr & Ruterjans (2000) J Magn Reson 146, 126-131). We performed tests including a systematic error underestimating the coupling constants by 5% for both NC^γ and C'C^γ ³J couplings and this yielded an error of less than 10 % in the estimated populations of the major rotamer states.

Point 9: *“The rotamer population derived from the 3J coupling analysis is quite different compared to the MD simulations, which mostly indicate one major rotamer – this should be discussed and clarified.”*

MD simulations that are carried out for 300, 500 or 1000 ns cannot be quantitatively compared to the NMR experiments results which average the coupling constant during the effective time period of the experiment when the J couplings are active; generally, tens of milliseconds. The MD experiments gave insight on the interactions, which cause the aromatics orientations to influence one another. We now discuss these points in the results section.

Point 10: *“The authors propose that the RNA bound state remains somewhat dynamic – what is the evidence for this? If so, how can the authors rule out that this dynamic features affect the structural calculation employed?”*

In the bound state, the recorded scalar couplings have intermediate values which indicate that rotamer averaging is present and therefore that the aromatic side-chains are dynamic even in the complexes. The structure determination is based on NOEs which have a 1/r⁶ distance dependence and therefore distance constraints more strongly weight short distances present in a dynamic ensemble. The conformers generated by the structure calculation protocol satisfy simultaneously as many NOEs as possible. It may not reflect the distribution of conformations of the aromatic side-chains. Thus, we needed the complementary χ1 experiments to obtain more information about the dynamics of the aromatic residues. Notably, the major populations determined from the J coupling data are in agreement with the NOE derived structures.

Point 11: *“Figures are somewhat complicated and would benefit from improving clarity.”*

We spent a considerable amount of time trying to simplify the figures given the complexity of the data we have to present and we reached a point that was satisfactory to us. We would nevertheless be happy to implement suggestions that the reviewer might have.

Point 12: *“The authors mention that the C-terminus transiently contacts the aromatic interface and might contribute to preference for gauche+ conformation in case of F426 (due to steric hindrance). Also, its deletion in MD simulation, alters the UP/DOWN conformation transition rates. It would be interesting to see if this can be corroborated by recording NMR scalar coupling experiments on this deletion mutant.”*

Additional experimental data would be interesting to carry out but the production and purification of the deletion mutant, which may not express and can potentially be unstable can be very time consuming and could considerably delay the present publication. For the coherence of the present work, we no longer discuss this observation.

Point 13: *“It is stated at 1 protein: 3 RNA ratio, 96% protein population is in bound form. This is probably extracted from the NMR titration, but it should maybe be mentioned in the Methods section.”*

We now mention in the methods section that the saturation of the protein was calculated based on the K_d determined by titration monitored with NMR.

Point 14: *“The NMR structural statistics reported in Table S1 under energy statistics (distance restraint violation and AMBER violation energy) seems not excellent. Distance violations seem rather high? Maybe somehow the numbers reported are the absolute no. of restraints violated and not mean \pm S.D.?”*

In the NMR statistics table, we report the average number of distance constraints violations \pm S.D. for different distance groups. On average, 0.6 restraints are violated by more than 0.4 Å in the ensemble, which is a relatively low value that denotes a good agreement of the structure with the restraints. The average and standard deviation of the maximal distance violation for the 20 conformers is 0.41 ± 0.06 Å. In this table, we do not report the average violation per constraint. Concerning the AMBER violation energy, it is in agreement with the distance restraints violation statistics.

Point 15: *“Figure legends of S3/4/5 are all mixed up. This should be corrected.”*

The order of the legends has been corrected.

Point 16: *“The numbering of residues is wrong in the text when compared with the Table1 (for example, residue K395A in Table is mentioned as K495 in text!).”*

We corrected the numbering in the table so it matches the text.

Point 17: *“In the Results section, the authors mention that they have 2325 intra-protein distance restraints while in the Table S1 it says 2347.”*

“We derived 2325 intra-protein distance restraints from NOESY experiments” refers to the total number of NOEs. The total number of intra-protein distance restraints is 2347. We now change the NMR statistic table to clarify this information.

Reviewer #3 (Remarks to the Author):

General remarks

Point 1: *“One concern is if the increase of affinity is due to a change in the binding mode (different*

nucleotides are recognized when the protein is mutated) and not only due to changes in the populated rotamers and most importantly, which is the biological relevance of studying the impact of these mutations in the affinity of the RNA recognition. Are some of these mutations observed in patients?"

The titrations of the mutated RNAs and the mutated RRM3s show chemical shift changes that are very similar for all the complexes. This indicates that the complexes all have a conserved binding mode and that the differences in chemical shift amplitudes reflect changes in affinity. Mutations in the non-canonical aromatic mutations have not been reported in diseases. This is not surprising. Selective pressure on the RRM3s ensures retention of the function. The non-canonical aromatic mutants are mechanistically important, as they were key for RRM3s ability to discriminate RNA targets and their relevance *in cell* is supported by the results of our luciferase reporter gene assay (Reviewer #2 General Point 2).

Point 2: *"One concern is that the authors described that the side chains orientations at the binding interface resemble that of the unbound protein. Since the affinity is low perhaps the bound rotamers are barely detectable. Is this a possibility?"*

The NMR titration of RRM3 in complex with 5'-UUUAA-3' shows that the protein is saturated at a protein:RNA ratio of 1:3. The restraints used to determine the structure of the protein and the scalar coupling data were obtained from experiments recorded at this ratio. Thus, our data reflect the main conformation in the bound form. Moreover, in the NOESYs, we could not detect NOEs that would be specific to the DOWN conformation indicating that the UP conformation is the major one. This is also supported by the scalar coupling data.

Point 3: *"Another concern is that the NOEs were automated assigned for the protein. Were the aromatic residues manually assigned? Were some of these Chemical shifts overlapped and ambiguously assigned? Perhaps the authors can provide additional information on how all these interactions and the rotamers of the residues were determined."*

The resonances of the aromatic residues were unambiguously assigned with the exception of HIS 429 CE1 and HE1 and PHE 455 CZ and HZ atoms that were not observable.

In NOESY spectra, we manually assigned NOEs originating from the aromatics and derived non-ambiguous upper limit distance restraints. These manually determined restraints were included as input during the initial automated peak picking, NOE assignment, and structure calculation performed by the ATNOS/CANDID algorithm together with CYANA. This procedure was used to ensure the picking of all peaks by ATNOS/CANDID. The resulting peak lists were cleared of assignments generated by ATNOS/CANDID and furnished as input for the macro NOEASSIGN in CYANA, which performed automated NOE assignment of the full protein without including the manual NOE assignment information.

Point 4: *"This is a common feature in the field of structural biology. The authors have made a deep study to characterize one example, that of the RRM domains."*

Our study reports not only on the solution structure of CUG-BP2 RRM3 in complex with 5'-UUUAA-3' and its aromatic side-chain rearrangement but also identifies similar rearrangements in a survey of RRM structures deposited in the PDB. RRM3s are the most abundant RNA binding domains hence understanding their recognition mode and deciphering general rules is fundamental to understand the

function of many RNA binding proteins. We now provide additional experimental data with our experiments on hnRNPA1 RRM1 (Reviewer #2 Specific comments Point 6) to support this idea.

Point 5: *“Including the structures characterized in this work (the pdb files, the restraints used to determine them and the measurements that define the rotamers described in the paper) would contribute to a better understanding of the described results.”*

The PDB files and the experimental restraints have been deposited in the PDB and the BMRB. They can be accessed under the following ID numbers: 5M8I (PDB) and 34057 (BMRB). Scalar couplings and rotamer populations are included in a table in the supplementary information.

Point 6: *“It will also explain the ensemble of conformations present in these structures (or the relative population of the conformers observed experimentally in the wild type). It is a bit contradictory that the presence of multiple rotamers explains the selectivity of the protein and at the same time, the interactions -even for a low affinity binding site-, are well-defined, allowing the measurement of hydrogen bonds with the RNA. Were some of these distances included as restraints during the calculation?”*

NMR structure calculation is driven by the NOEs. The ensemble satisfies as many of them at once as possible. Therefore, our structure describes the main conformation that yielded the observed NOEs. We have added text to clarify how the information obtained from the NOESYs and the experiments used to describe Chi1 angle distributions can be reconciled. The hydrogen bonds were geometrically defined (see supplementary table S2) and identified using the software MOLMOL. We did not include any hydrogen-bond restraints in our structure calculations.

Point 7: ***“On a more subjective note, do you feel that the paper will influence thinking in the field? Perhaps this influence might be very specific to the field of RRM domain interactions.”***

Please see our reply to point 1.

Specific comments:

Point 1: *“Line 80, high degree of homology. It should say high degree of similarity (or identity).”*

The wording was corrected.

Point 2: *“In several instances in the text, configuration is used instead of conformation.”*

The use of the word configuration was deliberate; we refer to the arrangement multiple moieties relative to one another. In the instances where it could be replaced by conformation, we have now changed it.

Point 3: *“ITC values need error boundaries.”*

We provided the standard deviation of our ITC values calculated on three independent measurements. We are not sure what the reviewer means by error boundaries.

REVIEWERS' COMMENTS:

Reviewer #2 (Remarks to the Author):

The authors have significantly revised the manuscript and added additional experiments also to support the role of RRM in RNA binding and function. The study still focusses on a rather specific and not abundant aspect or RRM-RNA recognition, that is likely not relevant for most RRMs (considering 3-10% of RRMs "may" also exhibit this as reported by the authors in the new Table S5), but now presents a more reasonable structure/function story.

Specific comments:

- The luciferase assay data seem to indicate a role of RRM for the function monitored, however, the effects related to changes in RRM3 binding interface (Try428 mutation) appear not statistically significant. The authors should thus tone down and more cautiously discuss any functional significance for the UP/DOWN binding mode.
- Table S5: please provide the 6/22 structures identified.
- The authors should make the J-coupling data also freely available at BMRB.

Reviewer #3 (Remarks to the Author):

Answers to the authors:

The authors have made an effort to clarify the information described in the paper and to answer the questions raised by the reviewers and editors. However, some of the replies are not fully satisfactory, for instance, the functional relevance of the mutations is still unclear. In addition, although the structural part of the manuscript is solid, the sections describing the Alanine mutations are confusing. I believe that they do not add relevant information to the main results of the work. If they add information, this information is restricted to scenarios that are originated by the presence of the Alanine substitutions.

I understand that the mutants might have been designed to demonstrate the role of the aromatic residues in the RNA interactions and/or to elucidate the mechanisms of binding and selectivity. If this was the aim, to clearly illustrate the mechanism/s, I think that the authors should have determined the structure of one or several Ala mutated proteins in complex with the UUUAA RNA. In my opinion, the main finding of their work is the binding mode they found for the recognition of the UUUAA site. This binding mode is similar to the unbound state of the domain and different from that of the previous work described by Tsuda et al and from many other complexes with other RRM domains. The comparison of both binding modes reveals the novelty of the present work.

A shortened version of the manuscript, focused on the determined structure and including a reduced section on the Ala mutations will simplify the message to the readers, focussing on native results and less on contacts or ring rearrangements that might or might not occur in a native context.

Reviewer #2 (Remarks to the Author):

The authors have significantly revised the manuscript and added additional experiments also to support the role of RRM in RNA binding and function. The study still focusses on a rather specific and not abundant aspect or RRM-RNA recognition, that is likely not relevant for most RRMs (considering 3-10% of RRMs “may” also exhibit this as reported by the authors in the new Table S5), but now presents a more reasonable structure/function story.

We are pleased that the revised manuscript and additional experiments were positively received by the reviewer #2. Regarding the specificity of our study, the statistics we present only cover RRMs for which structural data are available. Moreover, very little information exists on side-chain dynamics at the binding interface. Our study unveils an aspect of RRM-RNA recognition that may concern more RRMs than it is possible to estimate with the current state of the knowledge.

Specific comments:

- *The luciferase assay data seem to indicate a role of RRM for the function monitored, however, the effects related to changes in RRM3 binding interface (Try428 mutation) appear not statistically significant. The authors should thus tone down and more cautiously discuss any functional significance for the UP/DOWN binding mode.*

As suggested by the reviewer, we now moderate our conclusions

- *Table S5: please provide the 6/22 structures identified.*

We now provide the list of the six RRMs with multiple aromatic side-chain conformations and the twenty-two RRMs with aromatic side-chain in UP or mixed state conformation in an additional table.

- *The authors should make the J-coupling data also freely available at BMRB.*

We have now deposited the J-coupling data in the BMRB and they are available under the accession numbers 27140, 27154, 27142, 27155, 27152, 27156, 27153, 27157, 27163 and 27164.

Reviewer #3 (Remarks to the Author):

The authors have made an effort to clarify the information described in the paper and to answer the questions raised by the reviewers and editors. However, some of the replies are not fully satisfactory, for instance, the functional relevance of the mutations is still unclear. In addition, although the structural part of the manuscript is solid, the sections describing the Alanine mutations are confusing. I believe that they do not add relevant information to the main results of the work. If they add information, this information is restricted to scenarios that are originated by the presence of the Alanine substitutions. I understand that the mutants might have been

designed to demonstrate the role of the aromatic residues in the RNA interactions and/or to elucidate the mechanisms of binding and selectivity. If this was the aim, to clearly illustrate the mechanism/s, I think that the authors should have determined the structure of one or several Ala mutated proteins in complex with the UUUAA RNA.

In this study, the alanine scanning mutagenesis is used primarily to determine the contribution of specific residues to the RNA binding and gain mechanistic insights into the RNA discrimination mechanism of CUG-BP2 RRM3. We think that the combined scalar coupling data and MD simulations provide convincing evidence to support our hypothesis that aromatic side-chain rearrangement at the binding surface is key to modulate the interaction with RNA targets. However, solving the structure an alanine mutant in complex with the UUUAA RNA as suggested by the reviewer #3 is most certainly interesting for a follow up study.

In my opinion, the main finding of their work is the binding mode they found for the recognition of the UUUAA site. This binding mode is similar to the unbound state of the domain and different from that of the previous work described by Tsuda et al and from many other complexes with other RRM domains. The comparison of both binding modes reveals the novelty of the present work.

A shortened version of the manuscript, focused on the determined structure and including a reduced section on the Ala mutations will simplify the message to the readers, focussing on native results and less on contacts or ring rearrangements that might or might not occur in a native context.

As suggested by the reviewer, we have now reorganized and shorten the text. Instead of three paragraph, one on the alanine mutants affinities determination, one on the measurement of coupling constant and determination of rotamer population and one on the molecular dynamics simulation, we now have only two paragraphs. The first one focuses on the dynamics of aromatic side-chains at the surface of the wild type RRM3 and the second on how side-chain conformation at the binding surface is correlated to the gain or loss in affinity.